# T-SHIRT: Token-Selective Hierarchical Data Selection for Instruction Tuning

**Yanjun Fu    Faisal Hamman    Sanghamitra Dutta**
University of Maryland, College Park
{yanjunfu, fhamman, sanghamd}@umd.edu

## Abstract

Instruction tuning is essential for Large Language Models (LLMs) to effectively follow user instructions. To improve training efficiency and reduce data redundancy, recent works use LLM-based scoring functions, e.g., Instruction-Following Difficulty (IFD), to select high-quality instruction-tuning data with scores above a threshold. While these data selection methods often lead to models that can match or even exceed the performance of models trained on the full datasets, we identify two key limitations: (i) they assess quality at the sample level, ignoring token-level informativeness; and (ii) they overlook the robustness of the scoring method, often selecting a sample due to superficial lexical features instead of its true quality. In this work, we propose **T**oken-**S**elective **HI**e**R**archical Data Selection for Instruction **T**uning (T-SHIRT), a novel data selection framework that introduces a new scoring method to include only informative tokens in quality evaluation and also promotes robust and reliable samples whose neighbors also show high quality with less local inconsistencies. We demonstrate that models instruction-tuned on a curated dataset (only 5% of the original size) using T-SHIRT can outperform those trained on the entire large-scale dataset by up to 5.48 points on average across eight benchmarks. Across various LLMs and training set scales, our method consistently surpasses existing state-of-the-art data selection techniques, while also remaining both cost-effective and highly efficient. For instance, by using GPT-2 for score computation, we are able to process a dataset of 52k samples in 40 minutes on a single GPU. Our code is available at https://github.com/Dynamite321/T-SHIRT.

## 1 Introduction

Large Language Models (LLMs) have shown remarkable success in various tasks [1–5]. One key component of adapting these powerful LLMs for practical use is instruction tuning (also known as supervised fine-tuning, SFT) [6] that fine-tunes pre-trained LLMs on instruction–response pairs. During this stage, researchers often use millions of samples [3, 4, 7]. However, LIMA [8] challenges this norm with the Superficial Alignment Hypothesis, which indicates that *instruction tuning might not need to be data-intensive, shifting the focus from data quantity to data quality*. Fine-tuned on just 1,000 manually selected instruction-tuning samples, LIMA achieves strong performance. Data selection for instruction tuning offers several key benefits. First, full-dataset tuning is extremely time-consuming and costly. Furthermore, as synthesized data becomes more common [9–12], datasets often include redundancy and noise, making data selection for instruction tuning absolutely essential.

To select instruction tuning data, it is common to use scoring functions to assess sample quality and only keep samples with high scores. LIMA [8] relies on manual annotation, while others [13–15] prompt proprietary LLMs via APIs to assign quality scores. However, both methods are costly. An alternative is Instruction-Following Difficulty (IFD) score [16, 17] (elaborated in Equation (1)), which measures the ratio of a response's perplexity when conditioned on the instruction to its unconditioned

39th Conference on Neural Information Processing Systems (NeurIPS 2025).

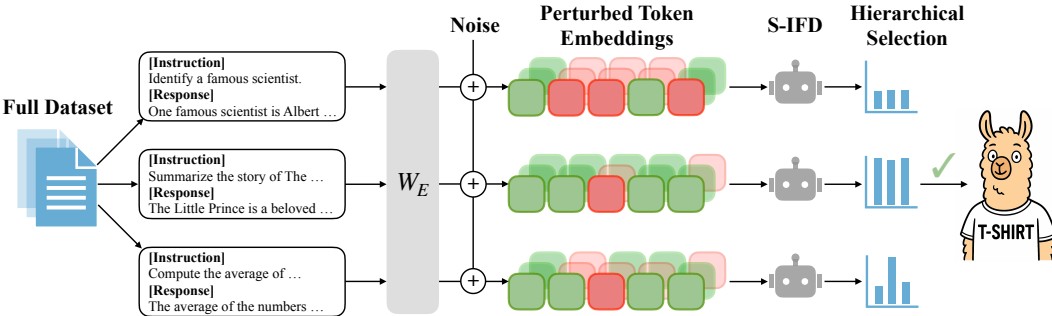

Figure 1: **An overview of our approach, T-SHIRT.** $W_E$ denotes the model's embedding layer (e.g., GPT-2) used to compute S-IFD scores. For each instruction-response pair, we generate its neighbors by perturbing token embeddings. Then we only use selected response tokens (green squares) for S-IFD computation, while excluded tokens are marked in red squares. Finally, we use hierarchical selection to choose samples whose neighbors exhibit high average S-IFD and low variance.

perplexity. Essentially, higher IFD scores indicate better quality. IFD is quite appealing due to its efficiency and low cost, and even small models like GPT-2 [18] can be used to compute it.

However, current scoring functions have multiple limitations as we identify in this work. *First, they evaluate the training data at the sample level, overlooking finer-grained token-level information.* Recent studies show that not all tokens contribute equally during training [19, 20]. Instruction tuning significantly affects models' output on only a small subset of response tokens [21, 22]. These findings motivate us to reconsider previous sample-level scoring paradigms. We revisit the computation of IFD scores to examine the informativeness of individual tokens. We find that the IFD score treats all response tokens equally, and a sample where none of the individual tokens are informative for instruction tuning may still receive a high overall score (see our counterexample in Figure 2). Empirically, we find that over 20% of the response tokens in instruction tuning datasets are not informative. Excluding these tokens from IFD calculation can substantially impact the final score.

*Second, existing works using LLM-based quality scores overlook the robustness of quality assessment, particularly with respect to the local neighborhood of a training sample.* These methods typically apply a fixed threshold and treat samples with scores above it as high quality. However, fluctuations may exist within a sample's neighborhood, where some of its neighboring samples fall below the threshold. We find that the IFD score is not a robust metric. Small, semantic-preserving changes, such as replacing a word with its synonym, can lead to large shifts in the IFD score (see our counterexample in Figure 3 with more details in Section 3.2). This suggests that high scores may also result from reliance on spurious features rather than reflecting true sample quality, motivating the need to also examine the local neighborhood of a sample during quality assessment.

To address the two challenges above, we propose **T-SHIRT**, short for **T**oken-**S**elective **HI**e**R**archical Data Selection for Instruction **T**uning, which is a new framework for data selection that promotes both token-level informativeness and robust quality assessment (also see Figure 1 for an overview). Towards introducing our framework, we first propose Selective IFD (S-IFD) (Equation (3)) that evaluates each response token individually and includes only informative tokens in the final quality score. Next, to avoid selecting samples whose high scores result from superficial lexical cues rather than true semantic quality, we go beyond previous pipelines that naively select samples based on a fixed quality score threshold. Instead, we apply a hierarchical selection strategy that favors samples whose neighbors exhibit high average and low variance in S-IFD scores. Experiments across multiple instruction tuning datasets and pretrained LLMs show that our proposed method T-SHIRT outperforms existing baselines on a wide range of downstream tasks. At the same time, our method remains computationally and financially efficient. For example, T-SHIRT requires only about 40 minutes to select data from the 52k-sample `Alpaca-GPT-4` [11] dataset using GPT-2 on a single GPU.

## 1.1 Related Work

**Data Selection for Instruction Tuning** LIMA [8] challenges the need for data-intensive instruction tuning [3, 4, 7] by proposing the Superficial Alignment Hypothesis (SAH) and showing that a 65B

model instruction-tuned on just 1,000 manually curated samples can outperform models trained with massive data. To reduce the cost of human annotation, subsequent research explored automated quality scoring functions for data selection. Early methods use simple metrics such as response length [23] and perplexity [24]. Later approaches rely on proprietary LLMs for quality scoring. For example, ALPAGASUS [13] and DS$^2$ [15] use OpenAI APIs, while DEITA [14] trains custom LLM scorers using ChatGPT-generated labels. To avoid expensive API calls, researchers propose using the IFD score [16], which can be computed even with small models like GPT-2 [17]. Other methods assess data quality via gradient-based influence functions [25] and Shapley values [26], but these approaches are computation-heavy and require additional validation sets. Besides sample quality, some works like LIMA [8], DEITA [14] and DS$^2$ [15], also emphasize sample diversity. However, most existing methods evaluate data quality at the sample level, overlooking token-level quality. They also do not consider the robustness of quality scores to small input variations.

**Token-level Informativeness** Recent works [21, 22] further support SAH by showing that alignment primarily affects models' outputs on a small subset of response tokens. This leads us to hypothesize that token-level analysis might be important for data selection. Other studies [19, 20] show that not all tokens contribute equally during pre-training and instruction tuning. Then they use selective language modeling (SLM) to alter the training process itself by computing losses only on selected tokens. Selecting tokens requires training a reference model on high-quality data and comparing token losses between the untrained and reference models. This makes data selection a prerequisite for building the reference model. In contrast, our metric, S-IFD, does not need a reference model and works well even if it is computed by GPT-2. Moreover, our method operates at the data preparation stage and does not change the training process. Thus, our method is orthogonal to SLM, and it has the potential to be combined with SLM for further improvement on instruction tuning.

**Robustness for LLMs** Theoretically, LLMs are not robust to minor input perturbations because the self-attention mechanism [27] is not smooth and Lipschitz continuous [28, 29]. Empirically, even small and semantic-preserving input changes can lead LLMs to misclassify text [30–32] or generate incorrect code [33]. Currently, robustness in LLM-based scoring functions for data selection remains underexplored. To the best of our knowledge, we are the first to incorporate robustness assessment into data selection for instruction tuning.

## 2 Preliminaries

**Problem Setting** Data selection for instruction tuning aims to identify the most informative subset from a large instruction-response dataset to efficiently fine-tune LLMs while maintaining strong performance on downstream tasks. Formally, given a dataset $\mathcal{D}$ containing $N$ instruction-response pairs and a selection budget $b < N$, a selection policy $\pi$ selects a subset $\mathcal{S} = \pi(\mathcal{D}, b)$, where $\mathcal{S} \subset \mathcal{D}$ and $|\mathcal{S}| = b$. Let $\theta$ denote the original LLM parameters and $\theta_{\mathcal{S}}$ the parameters after fine-tuning on $\mathcal{S}$. Under the budget $b$, the optimal selection policy $\pi^*$ maximizes the fine-tuned LLMs' performance: $\pi^* = \arg\max_\pi \phi(\theta_{\mathcal{S}})$, where $\phi$ is the performance metric, such as accuracy on test sets. Ideally, the policy $\pi$ should be both financially and computationally efficient to scale with large dataset sizes $N$.

**Scoring Function** Selection policies typically use scoring functions to evaluate each instruction-response pair in the dataset $\mathcal{D}$, favoring samples with higher scores. Some functions focus on a single aspect, such as quality or diversity of samples [13, 16, 17, 23], while others assess multiple attributes [8, 14, 15]. In this work, we focus solely on quality, using a well-established metric called Instruction-Following Difficulty (IFD) [16]. It is a desirable metric because it is purely statistical and easy to compute. For a given instruction-response pair $(x, y)$, the IFD score is computed using an LLM with parameters $\theta'$. It is defined as the ratio of the perplexity ($\text{PPL}_{\theta'}$) of the response conditioned on the instruction to the unconditional perplexity of the response:

$$\text{IFD}(x, y) = \frac{\text{PPL}_{\theta'}(y|x)}{\text{PPL}_{\theta'}(y)} = \frac{\exp\left\{-\frac{1}{T}\sum_{t=1}^{T} \log P_{\theta'}(y_t|y_{<t}, x)\right\}}{\exp\left\{-\frac{1}{T}\sum_{t=1}^{T} \log P_{\theta'}(y_t|y_{<t})\right\}}, \quad (1)$$

where $T$ is the length of the response, $y_t$ is the $t$-th token in the response, $y_{<t}$ represents all response tokens before the $t$-th token, and $P_{\theta'}$ is the probability assigned by the model $\theta'$ to the token $y_t$. A higher value of $P_{\theta'}$ indicates that the token $y_t$ is easy to predict under the model. The IFD score measures how much the instruction $x$ helps the model generate the response $y$. A lower score indicates that the instruction made the model easier to generate the response $y$, while a higher score

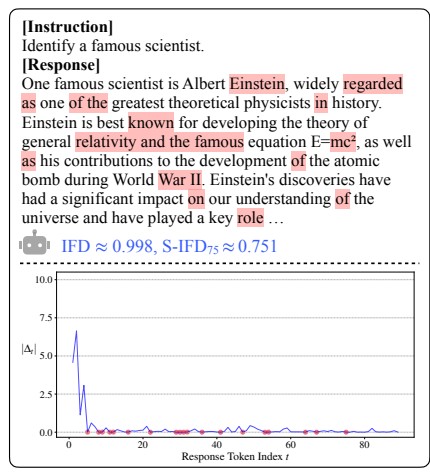
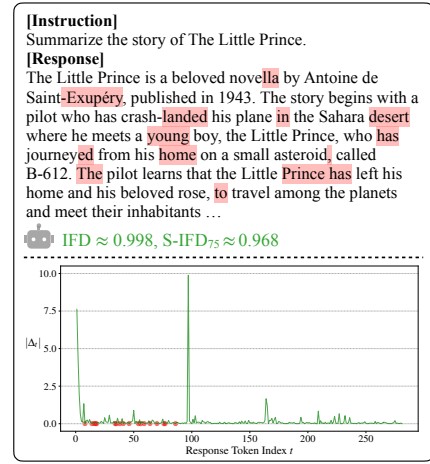

(a) Example 1: high IFD but low S-IFD$_{75}$.  (b) Example 2: high IFD and high S-IFD$_{75}$.

Figure 2: **Two examples from the** `Alpaca-GPT-4` **dataset with nearly identical IFD scores but markedly different S-IFD$_{75}$ scores. Top:** Instructions and partial responses from two examples, with tokens highlighted where $|\Delta_t| \leq 0.01$. **Bottom:** Plots of $|\Delta_t|$ values corresponding to the same examples. Tokens highlighted above are marked as red dots in the plots.

suggests greater difficulty and thus a more informative sample for fine-tuning. However, samples with $\text{IFD}(x, y) \geq 1$ are discarded, as they indicate that the instruction does not aid the response generation at all. The model used to compute IFD scores can be the same as the model being instruction-tuned $(\theta' = \theta)$ or a different one $(\theta' \neq \theta)$. SUPERFILTERING [17] shows that even small models like GPT-2 can be used for computing IFDs.

## 3 Main Contributions

### 3.1 Selective IFD to Capture Token-Level Informativeness

LLMs acquire most of their knowledge during pretraining. So even without the instruction, the token following a response prefix can be easy for them to predict. As shown in Figure 2a, when tokens "famous scientist" appear early in the response, pretrained LLMs can easily predict that the next token after "Albert" is "Einstein", regardless of whether the instruction is "Identify a famous scientist." or "Who proposed the theory of relativity?" This suggests that not all tokens are informative for instruction following. Therefore, we reconsider current data selection approaches, which typically evaluate instruction-response pairs as whole units without examining each token individually, and propose the question: "*Within an instruction-response pair, how informative is each response token?*"

To answer this, we revisit the IFD score. Using $\Delta_t$ to denote $\log P_{\theta'}(y_t|y_{<t}, x) - \log P_{\theta'}(y_t|y_{<t})$, we first rewrite Equation (1) as follows:

$$\text{IFD}(x, y) = \exp\left\{-\frac{1}{T}\sum_{t=1}^{T}\left(\log P_{\theta'}(y_t|y_{<t}, x) - \log P_{\theta'}(y_t|y_{<t})\right)\right\} = \exp\left\{-\frac{1}{T}\sum_{t=1}^{T}\Delta_t\right\}. \quad (2)$$

Here, $\Delta_t$ measures how much the log-likelihood of token $y_t$ changes after the instruction $x$ is provided. A small $|\Delta_t|$ indicates that the instruction makes little difference in generating the token, implying lower informativeness of $y_t$. Therefore, we propose $\Delta_t$ as a measure of token-level informativeness.

The IFD score treats all tokens equally, which might be misleading. We use the following thought experiment to demonstrate the potential issue intuitively. If for all $y_t$ in an instruction-response pair, $\Delta_t \approx 0.01$, the IFD score would be approximately 0.99, suggesting that this training sample has very high quality. However, at the token level, none of these tokens significantly depend on the instruction, indicating low informativeness. Empirically, we find that a large fraction of response tokens in instruction tuning datasets exhibit small $|\Delta_t|$ values, regardless of the choice of $\theta'$ for computing IFD scores. For instance, in the `Alpaca-GPT-4` dataset [11], using GPT-2 as $\theta'$, approximately 22% of

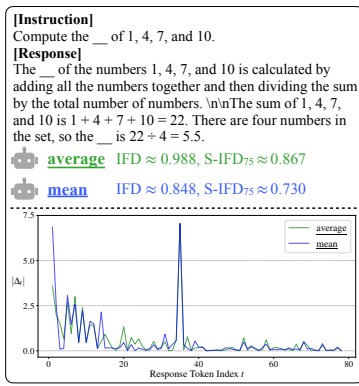

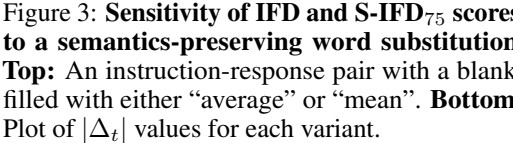

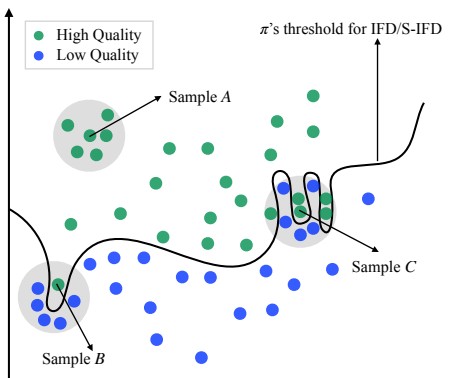

Figure 3: **Sensitivity of IFD and S-IFD$_{75}$ scores to a semantics-preserving word substitution. Top:** An instruction-response pair with a blank, filled with either "average" or "mean". **Bottom:** Plot of $|\Delta_t|$ values for each variant.

Figure 4: **Illustration of the selection policy $\pi$.** Each circle represents an instruction-response pair in the embedding space. Existing methods distinguish between high- and low-quality training data using a fixed threshold.

response tokens have $|\Delta_t| \leq 0.01$, as shown in Figure 5 of Appendix C.1. This proportion is around 28% for both Llama-3.1-8B [3] and Qwen-2.5-7B [4]. We further illustrate this by selecting two samples from the `Alpaca-GPT-4` dataset and computing token-level $|\Delta_t|$ using GPT-2. As shown at the bottom of Figure 2, we observe that $|\Delta_t|$ for most tokens is small.

**Our Approach** Building on our observation that not all tokens are equally informative, we propose Selective IFD (S-IFD), a refinement of the original IFD score that focuses only on the most informative tokens. Specifically, instead of treating all response tokens equally, S-IFD considers only those with the highest $|\Delta_t|$ values. To determine which tokens are informative, we introduce a fixed ratio $k\%$ and select the top $k\%$ of response tokens in the entire training dataset $\mathcal{D}$ based on their $|\Delta_t|$ rankings. Formally, S-IFD with a token selection ratio $k\%$ is defined as:

$$\text{S-IFD}_k\,(x,y) = \exp\left\{ -\frac{1}{\sum_{t=1}^{T} w_t} \sum_{t=1}^{T} w_t \Delta_t \right\},$$

$$\text{where } w_t = \begin{cases} 1 & \text{if } |\Delta_t| \text{ ranks top } k\% \text{ in the dataset } \mathcal{D}, \\ 0 & \text{otherwise.} \end{cases} \tag{3}$$

We illustrate the distinction between IFD and S-IFD using two examples in Figure 2, as they share similar IFD scores but exhibit notably different S-IFD values. We view that the second example, with a higher S-IFD score, has better quality.

## 3.2 Hierarchical Selection Based on Local Neighborhood

We note that LLMs can be highly sensitive to subtle input changes, even those that preserve semantics. As a result, for an instruction-response pair $(x, y)$, the computation of $\Delta_t$ for token $y_t$ may vary significantly with minor changes in the instruction $x$ or the preceding response tokens $y_{<t}$. These variations can accumulate, leading to substantial shifts in IFD and S-IFD scores. Even small perturbations to $(x, y)$ can lead to large fluctuations in IFD and S-IFD scores, as we illustrate through an example from the `Alpaca-GPT-4` dataset in Figure 3. Here, the instruction asks for the average of four integers. When the word "average" is replaced with the synonym "mean", both IFD and S-IFD$_{75}$ scores drop sharply. This highlights the model's sensitivity to superficial lexical variations and suggests that high scores may sometimes capture sensitivity to spurious surface features rather than genuine semantic quality.

The sensitivity of IFD and S-IFD scores to small input shifts motivates us to rethink these scores. Commonly, they regard training samples as high- or low-quality only based on whether their quality scores (e.g., IFD or S-IFD) exceed a given threshold [16, 17], which has several drawbacks as we illustrate in Figure 4. For example, sample $B$ may not be a reliable high-quality sample because many nearby points have low scores (essentially easy). This suggests that high average quality

among neighbors is desirable for a sample. However, average quality alone might be insufficient. For instance, sample $C$ lies in a region with high variance, indicating instability. In contrast, the neighbors of sample $A$ exhibit both high average scores and low local variance, making it a more robust candidate. Thus, we propose selecting samples based not only on high local average quality but also on low variance within their neighborhood.

**Our Approach** Motivated by our observation that local properties, including the average and variance of a sample's neighbors' S-IFD scores, are valuable for selection, we propose to use these measures as an additional dimension for evaluating data quality. To estimate these properties for a training sample $(x, y)$, we first generate its neighbors. Rather than using complex lexical perturbations such as paraphrasing or word substitution [31, 32], we opt for a simpler approach: injecting random noise into the token embeddings. We add uniformly distributed noise to each token embedding. For each token, we sample a noise vector with entries uniformly drawn i.i.d. from $[-1, 1]$, then scale it by $\epsilon = \alpha/\sqrt{(L+T)d}$, where $\alpha$ is a hyperparameter, $L$ and $T$ are the lengths of the instruction and response, respectively, and $d$ is the embedding dimension. This scaling results in perturbed samples whose expected $\ell_2$-norm distance from the original sample is $\alpha/\sqrt{3}$ [34]. We denote the perturbed embeddings of $(x, y)$ as $(x + \delta_x^{(i)}, y + \delta_y^{(i)})$, and compute the S-IFD score for each perturbed sample. We repeat this process $M$ times to estimate the average and variance of the S-IFD scores:[1]

$$
\begin{aligned}
\hat{\mu}(x, y) &= \frac{1}{M} \sum_{i=1}^{M} \text{S-IFD}_k\big(x + \delta_x^{(i)}, y + \delta_y^{(i)}\big), \\
\hat{\sigma}^2(x, y) &= \frac{1}{M} \sum_{i=1}^{M} \big(\text{S-IFD}_k\big(x + \delta_x^{(i)}, y + \delta_y^{(i)}\big) - \hat{\mu}(x, y)\big)^2,
\end{aligned}
\tag{4}
$$

where $\delta_x^{(i)} \sim \mathcal{U}^{L \times d}(-\epsilon, \epsilon)$, and $\delta_y^{(i)} \sim \mathcal{U}^{T \times d}(-\epsilon, \epsilon)$ with $\mathcal{U}$ denoting the uniform distribution.

After computing these local properties, we apply a hierarchical selection strategy to identify high-quality training samples. Given a budget $b$, we first select $\gamma b$ samples whose neighbors have the highest average S-IFD scores, where $\gamma > 1$ denotes an upsampling factor. From these, we further select the final $b$ samples with the lowest variance in S-IFD scores among their neighbors. We outline our data selection method in Algorithm 1 and visualize it in Figure 1. In practice, to improve efficiency, we batch the noisy embeddings instead of using a naive for-loop.

---

**Algorithm 1:** Token-Selective Hierarchical Data Selection for Instruction Tuning (T-SHIRT)

---

**Input:** Dataset $\mathcal{D}$, selection budget $b$, token selection ratio $k\%$, oversampling factor $\gamma$, base noise scale $\alpha$, and number of perturbations $M$

**foreach** $(x, y) \in \mathcal{D}$ **do**

    Compute $\epsilon \leftarrow \alpha/\sqrt{(L+T)d}$

    **for** $i \leftarrow 1$ **to** $M$ **do**

        Sample noise $\delta_x^{(i)} \sim \mathcal{U}^{L \times d}(-\epsilon, \epsilon)$, and $\delta_y^{(i)} \sim \mathcal{U}^{T \times d}(-\epsilon, \epsilon)$

        Compute perturbed embeddings $x' \leftarrow x + \delta_x^{(i)}$, $y' \leftarrow y + \delta_y^{(i)}$

        Compute S-IFD$_k(x', y')$ via Equation (3)

    Compute $\hat{\mu}(x, y)$ and $\hat{\sigma}^2(x, y)$ via Equation (4)

Select top $\gamma b$ samples from $\mathcal{D}$ with highest $\hat{\mu}(x, y)$ to construct $\hat{\mathcal{S}}$

From $\hat{\mathcal{S}}$, select final $b$ samples with lowest $\hat{\sigma}^2(x, y)$ to construct $\mathcal{S}$

**Output:** Selected subset $\mathcal{S} \subset \mathcal{D}$ of size $b$

---

# 4 Experiments

## 4.1 Experimental Setup

**Instruction Tuning Datasets** We conduct data selection on two widely used instruction-tuning datasets of different initial qualities and scales: `Alpaca-GPT-4` [11] and `Magpie` [10]. Following

---

[1]For simplicity, we use $\hat{\mu}(x, y)$ to denote $\mu(\text{S-IFD}_k(x, y))$, and $\hat{\sigma}^2(x, y)$ to denote $\sigma^2(\text{S-IFD}_k(x, y))$.

Table 1: **Performance comparison between our method, T-SHIRT, and other baseline methods on** `Alpaca-GPT-4`. We use each method to select 5% of the `Alpaca-GPT-4` training samples to instruction-tune Llama-3.1-8B and Qwen-2.5-7B. We use ARC-C for `ARC-Challenge`, HS for `HellaSwag`, TQA for `TruthfulQA`, GSM for `GSM8k`, AH for `Arena-Hard`, and AE-2 for `AlpacaEval` `2.0`. The best results are shown in **bold**, and the second-best are underlined. T-SHIRT achieves the best or comparable performance across all eight benchmarks and outperforms existing baselines in terms of $\mu_{\text{ALL}}$. Notably, it exceeds the full dataset performance by up to 5.48 points in $\mu_{\text{ALL}}$.

| | OpenLLM Leaderboard | | | | | | | LLM-as-a-Judge | | | $\mu_{\text{ALL}}$ |
|---|---|---|---|---|---|---|---|---|---|---|---|
| | ARC-C | HS | MMLU | TQA | BBH | GSM | $\mu_{\text{OPEN}}$ | AH | AE-2 | $\mu_{\text{LLM}}$ | |
| **Llama-3.1-8B** | | | | | | | | | | | |
| FULL | 55.97 | 77.89 | 59.77 | 53.32 | 43.86 | 32.75 | 53.93 | 5.20 | 9.10 | 7.15 | 42.23 |
| RANDOM | 57.94 | 81.29 | 61.37 | 53.39 | 45.62 | 30.71 | 55.05 | 5.50 | 8.13 | 6.82 | 42.99 |
| LONGEST | 58.45 | 83.07 | 61.83 | 54.86 | 46.40 | 51.25 | 59.31 | 6.80 | **11.91** | **9.36** | 46.82 |
| DEITA | 59.73 | 81.92 | 62.65 | 51.32 | 46.95 | 45.87 | 58.07 | 7.10 | 8.83 | 7.97 | 45.55 |
| DS$^2$ | 61.26 | 82.62 | 63.68 | 54.28 | **47.67** | 37.60 | 57.85 | **7.70** | 9.49 | 8.60 | 45.54 |
| IFD | 61.35 | 83.00 | 62.88 | 54.23 | 46.99 | 51.63 | 60.01 | 7.20 | 8.20 | 7.70 | 46.94 |
| T-SHIRT ($k = 50$) | 60.15 | **83.14** | 64.06 | **57.09** | 46.90 | 51.02 | 60.39 | 6.70 | 9.64 | 8.17 | 47.34 |
| T-SHIRT ($k = 75$) | **61.95** | 83.07 | **64.11** | 56.11 | 46.47 | **53.75** | **60.91** | 6.20 | 10.03 | 8.12 | **47.71** |
| **Qwen-2.5-7B** | | | | | | | | | | | |
| FULL | 61.69 | 79.54 | 72.19 | 58.37 | 50.82 | 77.10 | 66.62 | 11.50 | 8.81 | 10.16 | 52.50 |
| RANDOM | 64.68 | 81.97 | 74.27 | 59.06 | 53.34 | 33.13 | 61.08 | 14.70 | 16.45 | 15.58 | 49.70 |
| LONGEST | 65.61 | 82.47 | 73.92 | 61.48 | 53.52 | 84.61 | 70.27 | 14.30 | **19.15** | 16.73 | 56.88 |
| DEITA | 65.02 | 81.94 | 74.15 | 58.77 | 52.65 | 82.79 | 69.22 | 14.60 | 18.41 | 16.51 | 56.04 |
| DS$^2$ | 65.10 | 82.07 | **74.35** | 60.58 | 54.11 | 82.11 | 69.72 | 13.80 | 15.25 | 14.53 | 55.92 |
| IFD | 64.76 | **82.66** | 74.33 | 60.86 | 53.76 | 86.50 | 70.48 | 15.60 | 16.01 | 15.81 | 56.81 |
| T-SHIRT ($k = 50$) | **66.21** | 82.39 | 74.23 | **61.58** | 54.21 | 86.81 | **70.91** | 16.40 | 18.94 | 17.67 | **57.60** |
| T-SHIRT ($k = 75$) | 65.78 | 82.45 | 74.06 | 61.12 | 54.14 | **87.19** | 70.79 | 16.40 | 18.98 | 17.69 | 57.52 |

prior works [15–17], we use various data selection methods to select 5% of the 52k samples in `Alpaca-GPT-4` and approximately 3.3% (10k samples) of the 300k-sample `Magpie` dataset. These sampling ratios have been shown to reach an effective balance between model performance and data efficiency [15–17]. We provide additional details on these two datasets in Appendix B.1.

**Baselines** On `Alpaca-GPT-4`, we compare our method against the following baselines: (1) FULL, which uses the entire dataset for instruction tuning; (2) RANDOM, which randomly selects data from the full training set; (3) LONGEST [23], which selects samples with the longest response lengths; (4) DEITA [14], which first trains LLM scorers on ChatGPT-labeled data to assess sample quality and complexity, then selects data based on quality, complexity, and diversity; (5) DS$^2$ [15], which first uses GPT-4o-mini to score training samples, then learns a score transition matrix to correct potential scoring errors, and finally selects data based on both quality and diversity; (6) IFD [17], which uses GPT-2-computed IFD scores [16] to select high-quality samples. On `Magpie`, due to limited resources, we evaluate only the best-performing baselines identified on `Alpaca-GPT-4`. More details about the baselines are provided in Appendix B.2.

**Benchmarks** We evaluate the performance of the trained LLMs across eight diverse benchmarks. These include six standardized benchmarks from the OpenLLM Leaderboard [35, 36]: `ARC-Challenge` [37], `HellaSwag` [38], `MMLU` [39], `TruthfulQA` [40], `BBH` [41], and `GSM8k` [42]. For these benchmarks, we use the LM-Evaluation-Harness [43] and report their default evaluation metrics. We denote their average score as $\mu_{\text{OPEN}}$. In addition, we assess the instruction-tuned LLMs on two LLM-as-a-Judge benchmarks: `Arena-Hard` [44] and `AlpacaEval` `2.0` [45]. Following the recommendations of the respective benchmark creators, we report the style-controlled win rate [44] for `Arena-Hard` and the length-controlled win rate [45] for `AlpacaEval` `2.0`. The average score across these two is denoted as $\mu_{\text{LLM}}$. Finally, we report the overall average across all eight benchmarks as $\mu_{\text{ALL}}$. Additional details about the benchmarks are provided in Appendix B.3.

**Implementation Details of T-SHIRT** To implement our method, T-SHIRT, we use GPT-2 to compute S-IFD scores. For the hyperparameters in Algorithm 1, we set the token selection ratio $k\%$ to either 50% or 75%, the oversampling factor to $\gamma = 2$, the base noise scale to $\alpha = 5$ following

Table 2: **Performance comparison between our method, T-SHIRT, and other baseline methods on** `Magpie`**.** We use each method to select 10k samples from `Magpie` to instruction-tune Qwen-2.5-7B. We use abbreviated benchmark names as shown in Table 1. The best results are shown in **bold**, and the second-best are underlined. Our method achieves the best performance on seven out of eight benchmarks and has the highest $\mu_{\text{ALL}}$.

| | OpenLLM Leaderboard | | | | | | | LLM-as-a-Judge | | | $\mu_{\text{ALL}}$ |
|---|---|---|---|---|---|---|---|---|---|---|---|
| | ARC-C | HS | MMLU | TQA | BBH | GSM | $\mu_{\text{OPEN}}$ | AH | AE-2 | $\mu_{\text{LLM}}$ | |
| **Qwen-2.5-7B** | | | | | | | | | | | |
| LONGEST | 63.05 | **80.11** | 73.67 | 56.90 | 53.64 | 82.18 | 68.26 | 19.40 | 27.59 | 23.50 | 57.07 |
| IFD | **64.16** | 80.00 | 73.51 | 57.62 | 53.65 | 84.84 | 68.96 | 19.20 | 31.40 | 25.30 | 58.05 |
| T-SHIRT ($k = 50$) | **64.16** | 79.63 | **73.69** | **57.79** | **54.17** | **85.82** | **69.21** | **20.80** | **31.51** | **26.16** | **58.45** |

NEFTTUNE [34], and the number of perturbations to $M = 30$. We include more implementation details in Appendix B.4.

**Instruction Tuning Details** We use selected data to fine-tune two open-source LLMs, Llama-3.1-8B [3] and Qwen-2.5-7B [4]. Following the same training settings as prior works [16, 17, 46], we use a learning rate of 2e-5 and train for 3 epochs. For data selected from `Magpie`, we instead follow the setup in `Magpie` [10] and train for 2 epochs. Additional training details are provided in Appendix B.5.

### 4.2 Main Experimental Results

In this section, we present the experimental results to demonstrate the effectiveness of T-SHIRT. Additional results are provided in Appendix C.

**T-SHIRT outperforms all baselines.** We evaluate the performance of our method, T-SHIRT, and six baseline methods on the `Alpaca-GPT-4` dataset using eight benchmarks, with results presented in Table 1. Across two base models, T-SHIRT with different token-selection ratios ($k\%$) consistently outperforms all baselines in terms of $\mu_{\text{ALL}}$. Our data selection method achieves the highest performance on both the OpenLLM Leaderboard and LLM-as-a-Judge benchmarks. Specifically, T-SHIRT achieves the highest $\mu_{\text{OPEN}}$ and $\mu_{\text{LLM}}$ scores on Qwen-2.5-7B, as well as the highest $\mu_{\text{OPEN}}$ and competitive $\mu_{\text{LLM}}$ performance on Llama-3.1-8B. Compared to the strongest baseline, IFD, T-SHIRT delivers superior results on six out of eight benchmarks and performs comparably on the remaining two. These results demonstrate that our method significantly enhances the overall quality of selected data. Notably, selecting only 5% of the data, T-SHIRT improves $\mu_{\text{ALL}}$ by 5.10 to 5.48 points compared to instruction tuning using the full dataset.

**T-SHIRT is effective at selecting data across datasets of varying initial qualities and scales.** We further use T-SHIRT, and the two best-performing baselines from Table 1 to select 10k samples from `Magpie`, a recent state-of-the-art instruction tuning dataset. `Magpie` is approximately $5.8\times$ larger than `Alpaca-GPT-4` and has undergone multi-stage filtering to select 300k samples from an initial pool of one million, leading to higher initial quality. We use the selected data to fine-tune Qwen-2.5-7B and report results in Table 2. Across eight benchmarks, T-SHIRT outperforms the baselines on seven of them and achieves the best overall scores in $\mu_{\text{OPEN}}$, $\mu_{\text{LLM}}$, and $\mu_{\text{ALL}}$. These results demonstrate that T-SHIRT scales effectively and remains beneficial even on large datasets with high initial quality. We also fine-tune Qwen-2.5-14B using T-SHIRT-selected samples from `Magpie` and show detailed results in Table 5 and Appendix C.2.

**T-SHIRT is both cost-effective and computationally efficient.** When having strong performance, T-SHIRT relies only on a lightweight GPT-2 model to compute S-IFD scores, avoiding the need for API credits required by DEITA and DS$^2$. This makes our method both accessible and efficient. We present runtime comparisons in Table 3. On a single NVIDIA A6000 GPU, T-SHIRT runs 2.7 to $3.7\times$ faster than DEITA and DS$^2$. Although it is slower than LONGEST and IFD, its total runtime remains about 40 minutes, which is an acceptable trade-off given its superior overall performance. Moreover, we can further improve its efficiency by reducing the number of perturbations $M$ without hurting the performance (see ablation study on $M$ in Section 4.3).

**Proprietary LLMs are not (all) you need for data selection.** As shown in Table 1, the baselines LONGEST, IFD, and our method T-SHIRT outperform both DEITA and DS$^2$. LONGEST uses a

Table 3: **Runtime (in hours) for data selection methods on** `Alpaca-GPT-4`**.** We report total time including CPU, GPU, and API query time. All methods are run using a single GPU. For DEITA, we do not count the time for scoring data via API and training LLM-based scorers on scored data. T-SHIRT completes data selection in just 40 minutes, making it significantly faster than both DEITA and DS$^2$.

|  | Runtime (h) |
| --- | --- |
| LONGEST | 0.0 |
| DEITA | 1.9 |
| DS$^2$ | 2.6 |
| IFD | 0.2 |
| T-SHIRT ($M = 20$) | 0.5 |
| T-SHIRT ($M = 30$) | 0.7 |

Table 4: **Ablation study evaluating the effects of S-IFD and hierarchical selection.** ✓ indicates that the component is used, while ✗ indicates it is not. Specifically, ✗ for S-IFD denotes the use of the original IFD as the quality score. Thus, the first row corresponds to the baseline IFD, and the last row corresponds to our method. The results show that both S-IFD and hierarchical selection contribute positively to data selection performance, and their combination yields the best results.

| S-IFD | Hierarchical Selection | $\mu_{\text{OPEN}}$ | |
| --- | --- | --- | --- |
|  |  | Llama-3.1-8B | Qwen-2.5-7B |
| ✗ | ✗ | 60.01 | 70.48 |
| ✓ | ✗ | 60.66 | 70.78 |
| ✗ | ✓ | 60.84 | 70.78 |
| ✓ | ✓ | **60.91** | **70.91** |

single heuristic: response length. IFD and T-SHIRT assess data quality using only a weak model, GPT-2. In contrast, DEITA and DS$^2$ rely on expensive proprietary LLMs and employ more complex data selection strategies. Moreover, data selected by DEITA and DS$^2$ might contain intrinsic biases, because both of them show consistently lower performance on `GSM8k`, and DEITA also underperforms on `TruthfulQA`, regardless of the base LLM used for instruction tuning.

## 4.3 Ablation Studies and Analysis

In this section, we conduct ablation studies using the `Alpaca-GPT-4` dataset and OpenLLM Leaderboard benchmarks. Additional details are provided in Appendix D.

**Ablation on S-IFD and Hierarchical Selection** In Table 4, we evaluate the contributions of S-IFD and hierarchical selection. The results show that, compared to the baseline IFD, both components significantly improve $\mu_{\text{OPEN}}$ of instruction-tuned LLMs, highlighting the importance of token-level informativeness and local quality consistency in data selection. Combining both factors, our method achieves the best overall performance.

**Ablation on Neighbor Variance in Hierarchical Selection** As shown in Figure 6 in Appendix D.1, $\mu_{\text{OPEN}}$ drops significantly, by 2.29 to 4.86 points, when hierarchical selection chooses samples whose neighbors have high variance in S-IFD. This further highlights the importance of choosing data whose neighbors display consistently high quality.

**Ablation on Token Selection Ratio** Figure 7a in Appendix D.2 shows how $\mu_{\text{OPEN}}$ of LLMs changes with different token selection ratios $k\%$. For Llama-3.1-8B, the optimal ratio is 75%, while for Qwen-2.5-7B, it is 50%. These results support our observation in Section 3.1 that over 20% of the tokens in the instruction tuning dataset are not informative for effective data selection.

**Ablation on Token Weights** In Equation (3), we assign a weight ($w_t$) of 1 to informative tokens and 0 to uninformative ones, effectively removing the latter from the quality score computation. In Table 7 (Appendix D.3), we compare this binary weighting scheme with alternatives that upweight informative tokens instead of fully discarding uninformative ones. Specifically, we experiment with informative-to-uninformative token weight ratios of 1.5:1 and 2:1. Our results show that the binary weighting scheme consistently outperforms these softer alternatives. This suggests that completely ignoring uninformative tokens yields a cleaner and more reliable signal for computing quality scores.

**Ablation on Number of Perturbations** Figure 7b in Appendix D.4 illustrates the impact of varying the number of perturbations $M$ used in hierarchical selection. We observe that $\mu_{\text{OPEN}}$ for Llama-3.1-8B reaches its highest value at $M = 10$, while $\mu_{\text{OPEN}}$ for Qwen-2.5-7B peaks at $M = 20$. Both models perform well at $M = 30$. This suggests that reducing $M$ to lower values can improve the efficiency of our method without compromising performance.

# 5 Discussion

## 5.1 Token Embedding Perturbations vs. Lexical Perturbations

In the hierarchical selection process of T-SHIRT, we choose to generate neighbors of each sample using continuous token embedding perturbations rather than lexical perturbations. This decision is motivated by several challenges associated with lexical perturbations.

**Utility** Rule-based lexical perturbations often struggle to preserve semantic equivalence and fluency. As a result, prior studies [30, 32] typically rely on costly human evaluations to assess the quality and utility of perturbed examples.

**Efficiency** Generating lexical variants by prompting large language models through APIs is both computationally and financially expensive. For example, perturbing each of the 52k Alpaca-GPT4 samples 30 times would require over 1.5 million API calls.

## 5.2 Limitations and Future Work

While our method shows promising results in data selection for instruction tuning, it has several limitations, which we outline along with directions for future work.

**Model Scale** Due to computational constraints, our experiments are limited to models in the 7B–14B parameter range, as is common in several published papers in instruction tuning [10, 15, 20]. We will apply our data selection method to larger models in future work.

**Data Security Considerations** Our approach focuses solely on identifying high-quality subsets of instruction tuning datasets rather than security concerns. Future work could extend our method to incorporate data selection strategies that explicitly focus on security in LLMs.

**Theoretical Foundations** Our hierarchical selection method is inspired by research in adversarial machine learning [47, 48], robustness measures [49, 50], and LLM robustness [29, 34]. Empirical results support its effectiveness. However, a deeper theoretical understanding of our approach for data selection is still lacking. In future work, we aim to establish a stronger theoretical foundation to better explain our method and potentially even improve its performance.

# 6 Conclusion

In this paper, we present Token-Selective Hierarchical Data Selection for Instruction Tuning (T-SHIRT), a framework for selecting high-quality subsets from instruction tuning datasets. Our method introduces S-IFD, a quality metric that selectively incorporates only informative response tokens into quality assessment, and a hierarchical selection strategy that favors samples surrounded by consistently high-quality neighbors. Extensive experiments across a range of base LLMs and instruction tuning datasets demonstrate that T-SHIRT is both effective and scalable, consistently outperforming existing baselines. At the same time, it remains cost-effective and computationally efficient. These results underscore the importance of fine-grained and robust quality evaluation for data selection.

## Acknowledgments and Disclosure of Funding

We thank Peng Cao, Xianhang Cheng, and Yuehui Qian for their valuable comments on this paper. This work was partially supported by NSF CAREER 2340006. F. Hamman was also supported by Ann G. Wylie Dissertation Fellowship. Y. Fu gratefully acknowledges support from the NeurIPS 2025 Travel Award.

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

## Appendix

## Contents

## A  Summary of Contributions

Our contributions in this paper are summarized as follows:

- We analyze existing instruction tuning datasets and find that over 20% of the response tokens are uninformative. Excluding these uninformative tokens from IFD score computation significantly alters the results (see Section 3.1).

- We investigate the robustness of IFD and observe that it is sensitive to small, semantics-preserving perturbations. Our findings suggest that a high IFD score sometimes reflects superficial features of the training sample rather than its intrinsic quality (see Section 3.2).

- To address these limitations, we present T-SHIRT, a novel data selection framework. We first introduce S-IFD, a new quality metric that incorporates token-level informativeness into assessment. In addition, we propose a robust, hierarchical selection pipeline that prioritizes samples with high average neighborhood quality and low variance (see Section 3).

- We conduct extensive experiments showing that our method achieves superior performance using no more than 5% of the full instruction tuning datasets. Our approach consistently outperforms prior data selection methods across varying dataset sizes and base LLMs. Moreover, it remains cost-effective and efficient: on a single GPU, it processes a 52k-sample dataset in approximately 40 minutes, with no additional API costs (see Section 4).

## B  Experimental Setup Details

### B.1  Instruction Tuning Datasets

In this paper, we perform data selection on two instruction tuning datasets, `Alpaca-GPT-4` and `Magpie`, and they have varying scales and quality:

- `Alpaca-GPT-4` [11] contains 52k instruction-response pairs. It keeps the original instructions from the `Alpaca` dataset [46], but regenerates the responses using GPT-4 [2] instead of GPT-3.5, resulting in improved response quality.

- `Magpie` [10], specifically the `Magpie-Pro-300K-Filtered` version,[2] is a fully synthetic dataset comprising 300k high-quality instruction-response pairs. It is constructed by prompting Llama-3-70B-Instruct to generate one million samples, followed by multi-stage filtering. Compared to `Alpaca-GPT-4`, `Magpie` is not only larger but also of much higher quality.

## B.2 Baselines

We detail the three baselines: LONGEST, DEITA, and DS$^2$, which are used in our experiments:

- LONGEST [23] applies a simple heuristic that selects the longest samples from the dataset, based on the assumption that longer responses tend to be more informative and useful for instruction tuning. We tokenize responses using the Llama-3.1-8B tokenizer and select those with the highest token counts.
- DEITA [14] follows a four-step selection process. It begins with a small seed dataset and evolves its samples to span different levels of quality and complexity, similar to WIZARDLM [12]. Then, it prompts ChatGPT to label these samples by quality and complexity. Using the labeled data, it trains its own LLM-based scorers. Finally, it selects high-scoring, diverse samples from the full dataset, skipping those similar to ones already chosen. In our experiments, we use their released scorers and codebase to select data.
- DS$^2$ [15] also follows a four-step selection process. It first scores the full dataset by prompting GPT-4o-mini. Then, it learns a score-transition matrix to correct possible scoring errors. Next, it estimates each sample's diversity using feature embeddings by measuring the average cosine similarity to its $K$-nearest neighbors. Finally, it selects samples with both high corrected quality scores and high diversity. In our experiments, we use `GPT-4o-mini-2024-07-18` and DS$^2$'s codebase to score and select the data.

## B.3 Benchmarks

We extensively evaluate instruction-tuned LLMs using eight benchmarks, including six standardized ones from the OpenLLM Leaderboard [35, 36]:

- `ARC-Challenge` [37]: a challenging subset of the multi-choice question answering benchmark `ARC`, consisting of 2,590 natural science questions.
- `HellaSwag` [38]: a commonsense inference benchmark with 10k test samples, designed to assess models' ability to complete sentences plausibly.
- `MMLU` [39]: a diverse question-answering benchmark covering 57 tasks (e.g., STEM, US history), used to measure models' world knowledge and problem-solving skills.
- `TruthfulQA` [40]: a benchmark of 817 questions focused on assessing models' truthfulness in the face of common misconceptions and false beliefs.
- `BBH` [41]: a collection of 23 challenging tasks from `BIG-Bench` [51], often requiring multi-step reasoning.
- `GSM8k` [42]: a test set of roughly 1k grade-school-level math word problems targeting multi-step mathematical reasoning.

Following the OpenLLM Leaderboard, we use the codebase of LM-Evaluation-Harness [43] for evaluation. We report normalized accuracy for `ARC-Challenge`, `HellaSwag`, and `BBH`; accuracy for `MMLU` and `TruthfulQA`; and exact match for `GSM8k`.

Then we evaluate the instruction-tuned LLMs using two LLM-as-a-Judge benchmarks, where strong LLMs act as proxies for human annotators by comparing responses from the evaluated model and a baseline model, then indicating a preference. Specifically, we use:

- `Arena-Hard` [44]: comprises 500 challenging queries sourced from `Chatbot Arena` [52] and `WildChat-1M` [53]. It introduces the style-controlled win rate (SC) to mitigate judges' bias related to stylistic features, including answer length and density of markdown headers. We use `GPT-4-0314` as the baseline model and `Gemini-2.5-Flash-Preview-04-17` as the judge, reporting SC scores.

---

[2] https://huggingface.co/datasets/Magpie-Align/Magpie-Pro-300K-Filtered

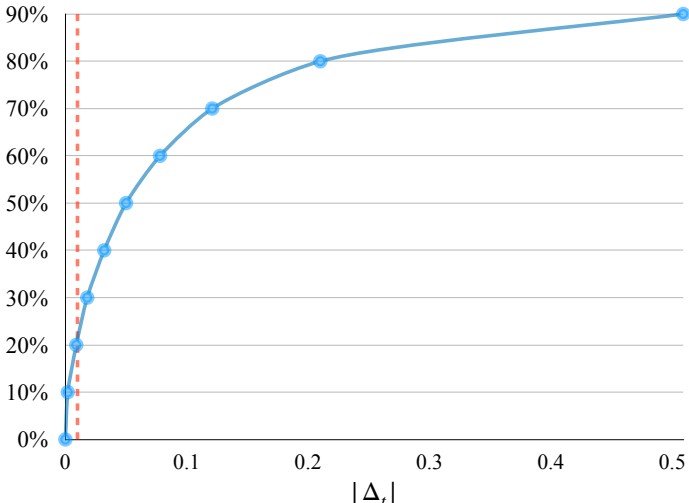

Figure 5: **CDF of $|\Delta_t|$ (Equation (2)) for response tokens in `Alpaca-GPT-4`, computed using GPT-2.** The red dash line indicates $|\Delta_t| = 0.01$.

- `AlpacaEval 2.0` [45]: contains 805 queries from `AlpacaEval` [46] and uses the length-controlled win rate (LC) to mitigate judges' bias on response length. We use `GPT-4-1106-Preview` as the baseline model and `GPT-4o-2024-08-06` as the judge, reporting LC scores.

Compared to the official settings of `Arena-Hard` and `AlpacaEval 2.0`, we use the same baseline models but different judges due to API budget constraints.

### B.4 Implementation Details of T-Shirt

To implement hierarchical selection in our method, a straightforward approach would be to iterate $M$ times and recompute the S-IFD score for each perturbed sample individually, as shown in Algorithm 1. However, this is computationally inefficient, especially since GPT-2 is relatively small and leaves much of the GPU unused. Instead, we generate a noise tensor of shape $M \times (L + T) \times d$, where $M$ is the number of perturbations, $L$ is the instruction length, $T$ is the response length, and $d$ is the embedding dimension. This noise is added to the sample's token embeddings using broadcasting. This allows us to compute perplexities for all $M$ perturbed samples in a single forward pass through GPT-2, which significantly reduces the time cost compared to repeated evaluations with a for-loop.

### B.5 Instruction Tuning Details

For data selected from `Alpaca-GPT-4`, we follow the training settings used in prior works [16, 17, 46]: a learning rate of 2e-5, 3 training epochs, a batch size of 128, a warmup ratio of 0.03, and a cosine learning rate schedule. For data selected from `Magpie`, we keep the same settings but reduce the number of training epochs to 2, in line with the official configuration of `Magpie` [10]. All instruction tuning experiments are conducted on a server equipped with NVIDIA A6000 GPUs.

## C Additional Experiments

### C.1 CDF of $|\Delta_t|$ with GPT-2 as $\theta'$

Figure 5 shows the cumulative distribution function (CDF) of $|\Delta_t|$ (as defined in Equation (2)) for response tokens in the `Alpaca-GPT-4` dataset, computed using GPT-2 as $\theta'$. Notably, 20% of tokens have $|\Delta_t| \leq 0.009$, and 50% have $|\Delta_t| \leq 0.050$, indicating that a substantial portion of tokens are relatively uninformative even for GPT-2.

Table 5: **Performance comparison between our method, T-SHIRT, and other baseline methods on Qwen-2.5-14B and** `Magpie`**.** We use each method to select 10k samples from `Magpie` to instruction-tune Qwen-2.5-14B. We use abbreviated benchmark names as shown in Table 1. The best results are shown in **bold**, and the second-best are underlined.

| | OpenLLM Leaderboard | | | | | | | LLM-as-a-Judge | | | $\mu_{\text{ALL}}$ |
|---|---|---|---|---|---|---|---|---|---|---|---|
| | ARC-C | HS | MMLU | TQA | BBH | GSM | $\mu_{\text{OPEN}}$ | AH[3] | AE-2 | $\mu_{\text{LLM}}$ | |
| **Qwen-2.5-14B** | | | | | | | | | | | |
| LONGEST | 68.17 | 84.01 | 79.23 | **58.69** | 61.00 | **85.22** | 72.72 | 34.40 | 35.01 | 34.71 | 63.22 |
| IFD | **68.94** | **84.05** | 79.07 | 57.87 | 61.78 | 83.02 | 72.46 | 33.10 | 37.15 | 35.13 | 63.12 |
| T-SHIRT ($k=50$) | 68.60 | 83.90 | **79.26** | 58.60 | 62.04 | 84.15 | **72.76** | **35.30** | **38.45** | **36.88** | **63.79** |

Table 6: **Average performance and standard deviation of RANDOM and T-SHIRT across three random seeds on Qwen-2.5-7B and** `Alpaca-GPT-4`**.** We use abbreviated benchmark names as shown in Table 1.

| | OpenLLM Leaderboard | | | | | | |
|---|---|---|---|---|---|---|---|
| | ARC-C | HS | MMLU | TQA | BBH | GSM | $\mu_{\text{OPEN}}$ |
| **Qwen-2.5-7B** | | | | | | | |
| RANDOM | $64.42_{\pm0.31}$ | $81.89_{\pm0.07}$ | $\mathbf{74.15}_{\pm0.11}$ | $59.30_{\pm0.18}$ | $53.46_{\pm0.09}$ | $50.52_{\pm14.17}$ | $63.96_{\pm2.32}$ |
| T-SHIRT ($k=50$) | $\mathbf{66.04}_{\pm0.24}$ | $\mathbf{82.36}_{\pm0.03}$ | $74.09_{\pm0.10}$ | $\mathbf{61.42}_{\pm0.17}$ | $\mathbf{54.21}_{\pm0.11}$ | $\mathbf{87.34}_{\pm0.59}$ | $\mathbf{70.91}_{\pm0.07}$ |

## C.2  Performance of T-SHIRT on Qwen-2.5-14B

In the main experiments (Section 4), we primarily evaluate T-SHIRT on models with 7B–8B parameters. Here, we extend the evaluation to Qwen-2.5-14B and compare T-SHIRT against LONGEST and IFD using experimental settings similar to those in Section 4.2. The results are shown in Table 5. On `Magpie`, T-SHIRT consistently outperforms both baselines across $\mu_{\text{OPEN}}$, $\mu_{\text{LLM}}$, and $\mu_{\text{ALL}}$. Notably, it achieves an average improvement of at least 1.75 points on the LLM-as-a-Judge benchmarks.

## C.3  Standard Deviation of Results

Both the baseline RANDOM and our proposed method T-SHIRT involve stochasticity. Within our limited budget, we run each method on Qwen-2.5-7B and `Alpaca-GPT-4` using three different random seeds, and report the mean and standard deviation of their performance on the OpenLLM leaderboard benchmarks. The results, shown in Table 6, are consistent with the observations reported in the main paper.

# D  Ablation Study Details

This section provides details of our ablation study, where we use the average performance on the OpenLLM Leaderboard benchmarks, $\mu_{\text{OPEN}}$, as the primary metric. For ease of reading, we make this section self-contained by including the discussion also provided in the main paper with additional details.

## D.1  Ablation on Neighbor Variance in Hierarchical Selection

We examine the role of selecting samples whose neighbors exhibit low quality variance in the hierarchical selection process (see Algorithm 1 and Section 3.2 for details). As shown in Figure 6, performance for both Llama-3.1-8B and Qwen-2.5-7B drops significantly when high-variance neighborhoods are preferred over low-variance ones. This highlights the importance of selecting samples from consistently high-quality neighborhoods and further validates the effectiveness of our hierarchical selection approach.

---

[3]While conducting this experiment, `Gemini-2.5-Flash-Preview-04-17` became unavailable, so we used `Gemini-2.5-Flash` instead.

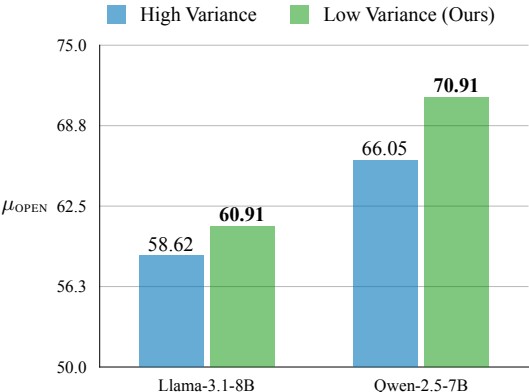

Figure 6: **Ablation study on the impact of preferring samples with low-variance** ($\hat{\sigma}^2(x, y)$ **in Equation (4)) versus high-variance neighbors during hierarchical selection.** The results indicate that selecting samples whose neighbors exhibit high variance significantly degrades performance.

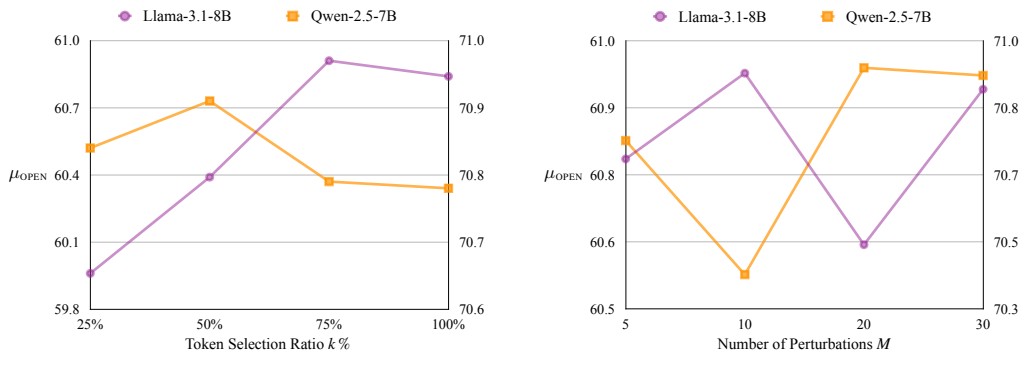

(a) Ablation study on token selection ratio $k\%$.  (b) Ablation study on number of perturbations $M$.

Figure 7: **Ablation study on two hyperparameters: the token selection ratio $k\%$ in S-IFD and the number of perturbations $M$ in hierarchical selection.** In both subfigures, the left y-axis represents $\mu_{\text{OPEN}}$ for Llama-3.1-8B, and the right y-axis represents $\mu_{\text{OPEN}}$ for Qwen-2.5-7B. (a) shows that the optimal token selection ratios are 75% for Llama-3.1-8B and 50% for Qwen-2.5-7B. (b) indicates that the optimal number of perturbations is 10 for Llama-3.1-8B and 20 for Qwen-2.5-7B. Both models maintain strong performance when $M = 30$.

Table 7: **Ablation study on token weights ($w_t$ in Equation (3)) for informative vs. uninformative tokens.** We use abbreviated benchmark names as defined in Table 1. The best results are shown in **bold**, and the second-best are underlined. The results indicate that completely removing uninformative tokens from the S-IFD computation is more effective than merely upweighting informative tokens.

| $w_t$ Ratio | OpenLLM Leaderboard | | | | | | |
|---|---|---|---|---|---|---|---|
| | ARC-C | HS | MMLU | TQA | BBH | GSM | $\mu_{\text{OPEN}}$ |
| **Qwen-2.5-7B** | | | | | | | |
| $1.5:1$ | 65.44 | 82.37 | 74.09 | 61.44 | 53.83 | 86.05 | 70.54 |
| $2:1$ | 65.53 | 82.17 | 74.19 | 60.73 | 53.95 | **86.81** | 70.56 |
| **$1:0$ (Ours)** | **66.21** | **82.39** | **74.23** | **61.58** | **54.21** | **86.81** | **70.91** |

## D.2 Ablation on Token Selection Ratio

We ablate the token selection ratio $k\%$ to assess its impact on data selection performance. Here, a ratio of 100% corresponds to using IFD (i.e., all tokens are used for computing quality scores), rather than S-IFD. As shown in Figure 7a, Llama-3.1-8B achieves optimal performance at $k = 75$, while Qwen-2.5-7B's performance peaks at $k = 50$. These results support our observation in Section 3.1

that over 20% of response tokens in instruction tuning datasets are uninformative. Using S-IFD to focus only on informative tokens enhances data quality assessment and improves data selection for instruction tuning.

### D.3   Ablation on Token Weights

We ablate the choice of token weights $w_t$ used in computing the S-IFD score in Equation (3). In addition to removing uninformative tokens by setting their weights to 0, we also experiment with upweighting informative tokens by setting the weight ratios between informative and uninformative tokens to 1.5:1 and 2:1. As shown in Table 7, removing uninformative tokens entirely yields better performance than merely upweighting informative ones.

### D.4   Ablation on Number of Perturbations

We ablate the effect of the number of perturbations $M$ in hierarchical selection. Llama-3.1-8B performs best at $M = 10$, while Qwen-2.5-7B reaches optimal performance at $M = 20$. Both models maintain strong performance at $M = 30$, suggesting that our method, T-SHIRT, can be made more efficient by reducing $M$ without sacrificing performance.

