# OpenReview forum: "T-SHIRT: Token-Selective Hierarchical Data Selection for Instruction Tuning"
_NeurIPS.cc/2025/Conference — NeurIPS 2025 poster_

### Official Review · Reviewer_VJnr · 2025-06-09

**Clarity:** 4
**Significance:** 3
**Originality:** 3
**Rating:** 5
**Confidence:** 4

**Summary:**

Selection of high quality data is critical for efficient and effective instruction-tuning. Existing methods turn toward sample-level quality scores assigned by humans or proprietary LLMs, which are expensive to compute and overlook token-level features of instruction samples. Further, semantically similar samples can often have varying quality score assignments, which is not accounted for by threshold-based data selection. This paper addresses these challenges through T-Shirt, a reference-free data selection framework that considers only informative tokens when assigning quality scores and prioritizes samples whose neighbors achieve high quality scores to ensure robustness and consistency. Empirical evaluations show T-Shirt is highly scalable and efficient while incurring low monetary and time costs.

**Questions:**

* How does random noise injection compares to systematic lexical perturbations when it comes to IFD and S-IFD scores, and selection efficacy?
* How effective is the pipeline toward fine-tuning LLMs across model scales (e.g., efficacy of the selected data for larger models trained with LoRA, will the observed performance gains still hold)?

**Ethical Concerns:**

["NO or VERY MINOR ethics concerns only"]

**Final Justification:**

The authors have addressed my concerns regarding scalability and hyperparameter selection as it relates to the method. It appears other reviewers' concerns are mostly resolved as well. I therefore maintain my positive rating of the paper.

**Limitations:**

Yes

**Quality:**

3

**Strengths And Weaknesses:**

Strengths
* The paper introduces a novel proposal to consider token-level information and local quality scores of related neighbor data points to evaluate training data quality.
* The proposed methodology is well-motivated and clearly described; figures and data presentation are effective.
* Experiments have good coverage of relevant benchmarks, baselines, and ablations, and results are strong even when using only GPT2 for score computation.
* The paper provides evidence that using even a weak selector model can get effective samples in short amount of time, consistent with some recent works.

Weaknesses
* No major weaknesses. While results are convincing, the T-Shirt-selected data is only used to fine-tune two moderately sized base LLMs (<=8B parameters), with efficacy at larger scales unclear.

---

> ### Author Rebuttal · Authors · 2025-07-30
>
> We thank the reviewer for the encouraging assessment on the technical novelty, efficiency, experimental thoroughness, and presentation quality of our work. We also appreciate the thoughtful questions and constructive suggestions. Below, we address the weakness (W) and the questions (Q) raised by the reviewer.
>
> ### **Notations and clarifications.**
> In our response, we follow the draft’s notations for several benchmark names: ARC-C (ARC-Challenge), HS (HellaSwag), TQA (TruthfulQA), GSM (GSM8k), AH (Arena-Hard), and AE-2 (AlpacaEval 2.0). We use $\mu_{\text{open}}$ to denote the average performance across six OpenLLM Leaderboard benchmarks, $\mu_{\text{LLM}}$ for the average performance across two LLM-as-a-Judge benchmarks, and $\mu_{\text{all}}$ for the average performance over all eight benchmarks. Additionally, we clarify that *Gemini-2.5-Flash-Preview-04-17* was used in our draft as the judge for evaluating AH. Since this checkpoint is no longer available via API, we now use the stable version of *Gemini-2.5-Flash*, which affects the absolute scale of AH scores.
>
> ### **W1 & Q2: Applicability to larger models.**
> We thank the reviewer for this constructive suggestion. To evaluate the scalability of our method, we fine-tuned a *Qwen-2.5-14B* model using data selected from the Magpie dataset, following the setup described in Table 2. Due to time constraints, we used a token-selection ratio of $k$% $= 50$% without further hyperparameter tuning. The results are presented below.
>
> | Method                      | ARC‑C |   HS   | MMLU |  TQA |  BBH |  GSM | $\mu_{\text{open}}$ |   AH  | AE‑2 | $\mu_{\text{LLM}}$ | $\mu_{\text{all}}$ |
> | :---------------------------|:-----: |:-----: |:-----: |:-----: |:-----: |:-----: |:-----: |:-----: |:-----: |:-----: |:-----: |
> | **Longest**                 | $68.17$ | $\underline{84.01}$ | $\underline{79.23}$ | $\mathbf{58.69}$ | $61.00$ | $\mathbf{85.22}$ | $\underline{72.72}$ | $\underline{34.40}$ | $35.01$ | $34.71$ | $\underline{63.22}$ |
> | **IFD**                     | $\mathbf{68.94}$ |	$\mathbf{84.05}$ | $79.07$ | $57.87$ | $\underline{61.78}$ | $83.02$ | $72.46$ | $33.10$ | $\underline{37.15}$ | $\underline{35.13}$ | $63.12$ |
> | **T‑SHIRT ($k=50$)**        | $\underline{68.60}$ | $83.90$ | $\mathbf{79.26}$ | $\underline{58.60}$ | $\mathbf{62.04}$ | $\underline{84.15}$ | $\mathbf{72.76}$ | $\mathbf{35.30}$ | $\mathbf{38.45}$ | $\mathbf{36.88}$ | $\mathbf{63.79}$ |
>
> T-SHIRT consistently outperforms the strongest baselines across $\mu_{\text{open}}$, $\mu_{\text{LLM}}$, and $\mu_{\text{all}}$. Notably, it achieves an average improvement of at least 1.75 points on the LLM-as-a-Judge benchmarks. These results suggest that our method remains effective and robust when applied to instruction tuning of larger models beyond the 7–8B scale.
>
> We will include these results in the revised version of the draft.
>
> ### **Q1: Comparison between random noise injection and lexical perturbations.**
> Compared to random noise injection at the embedding level, lexical perturbations present several challenges:
> * **Utility.** Rule-based lexical perturbations may fail to maintain semantic equivalence and fluency. Prior studies \[1, 2\] typically rely on expensive human evaluations to verify the utility of perturbed examples.
> * **Efficiency.** Generating lexical variants via prompting powerful LLMs through APIs is computationally and financially expensive. For instance, perturbing each of the 52k Alpaca-GPT4 samples 30 times would require over 1.5 million API calls.
>
> Given these limitations, we adopt random noise injection as a lightweight and scalable alternative to lexical perturbations. Investigating how systematic lexical perturbations can affect selection efficacy remains an exciting direction for future work.
>
> **References**
>
> \[1\] D. Jin, Z. Jin, J. T. Zhou, and P. Szolovits. Is BERT Really Robust? A Strong Baseline for Natural Language Attack on Text Classification and Entailment, 2020.
>
> \[2\] B. Wang, C. Xu, S. Wang, Z. Gan, Y. Cheng, J. Gao, A. H. Awadallah, and B. Li. Adversarial GLUE: A Multi-Task Benchmark for Robustness Evaluation of Language Models, 2021.

---

> > ### Comment · Reviewer_VJnr · 2025-08-01
> >
> > Dear authors, thank you for the response and clarifications. My concerns are addressed and I will maintain my positive assessment of the paper.

---

> > > ### Author Response · Authors · 2025-08-05
> > >
> > > Dear reviewer VJnr,
> > >
> > > Thank you for your positive assessment and for actively participating in the discussion! We are glad to hear that our response addressed your concerns. Your feedback greatly helped us strengthen our work, and we will incorporate the results and clarifications from our response into the revised version of the paper.

---

### Official Review · Reviewer_MFfG · 2025-06-21

**Clarity:** 4
**Significance:** 3
**Originality:** 3
**Rating:** 5
**Confidence:** 4

**Summary:**

This paper identifies two key limitations of prior data selection approaches (e.g., IFD) for instruction tuning: (i) they evaluate sample-level data quality, ignoring token-level information; and (ii) they overlook the robustness of data quality assessment. To address these issues, the authors propose T-SHIRT, a new data selection framework that incorporates fine-grained token-level information and favors samples with consistently high-quality neighbors. They conduct comprehensive experiments demonstrating that datasets curated using T-SHIRT outperform full datasets and those filtered by other data selection methods.

**Questions:**

- Have you considered other scoring models used in score computation? Do you have any insights into how the choice of scoring model influences the quality of data selection?

**Ethical Concerns:**

["NO or VERY MINOR ethics concerns only"]

**Final Justification:**

The author's reply about the token-level advantage of their method makes sense, though they do not provide direct evidence. The new ablation experiments on removing tokens solve my question well.

**Limitations:**

yes

**Paper Formatting Concerns:**

No major formatting issues.

**Quality:**

3

**Strengths And Weaknesses:**

### Strengths
- The authors use a semantics-preserving word substitution example to illustrate that IFD-based scores are highly sensitive to subtle textual changes, suggesting that selecting samples with high-quality neighbors is more effective than relying solely on fixed score thresholds.
- This study provides fine-grained signals for high-quality data selection by incorporating token-level information, emphasizing that not all tokens contribute equally when assessing example quality.
- They conduct comprehensive experiments using two instruction-tuning datasets of different scales, evaluate performance on eight benchmarks, and compare results against six baseline methods.
- The authors perform comprehensive ablation studies to evaluate the contribution of each major component in their proposed method.
### Weaknesses
- Since this study emphasizes token-level information when selecting training examples, it would be interesting to investigate whether examples selected by T-SHIRT offer advantages at the token level compared to those selected by other baselines during instruction tuning.
- I’m not fully convinced that removing all tokens with small delta_t values is justified. Since these values are already quite small, their cumulative contribution to the final score may be negligible. Could you please provide a plot showing the distribution of delta_t values to better understand their overall impact? Additionally, have you considered weighting tokens by their delta_t values instead of removing the less significant ones entirely?

---

> ### Author Rebuttal · Authors · 2025-07-30
>
> We thank the reviewer for the positive feedback on the technical novelty, comprehensive experiments, and ablation studies in our work. We also appreciate the reviewer’s insightful comments and questions on token-level informativeness and the choice of scoring models. Below, we provide itemized responses and hope they can address weaknesses (W) and questions (Q) raised by the reviewer.
>
> ### **Notations.**
> We use the following notations throughout our response, and they are consistent with our draft.
> * $\mathcal{D}$: instruction-tuning dataset;
> * $\theta$: LLM parameters to be fine-tuned;
> * $(x, y)$: instruction–response pair;
> * $y_t$: the $t$-th token in the response;
> * $|\Delta_t|$: token-level informativeness for the token $y_t$;
> * Benchmark abbreviations: ARC-C (ARC-Challenge), HS (HellaSwag), TQA (TruthfulQA), GSM (GSM8k);
> * $\mu_{\text{open}}$: average score over six OpenLLM Leaderboard benchmarks.
>
> ### **W1: Token-level advantages of T-SHIRT.**
> We appreciate the reviewer’s suggestion to investigate whether T-SHIRT offers token-level advantages during instruction tuning.
>
> **Tokenizer difference limits direct comparison.** The models used in our pipeline, *GPT-2* for scoring, and *Llama-3.1-8B* and *Qwen-2.5-7B* for fine-tuning, employ different tokenizers, and this makes direct token-level comparisons nontrivial. Therefore, we analyze the training dynamics as a proxy.
>
> **Higher training loss indicates more informative tokens.** We observe that LLMs fine-tuned on data selected by T-SHIRT consistently exhibit higher training loss compared to those trained on baseline-selected data. This pattern holds across both *Llama-3.1-8B* and *Qwen-2.5-7B*, and across datasets (Alpaca-GPT4 and Magpie). Since the instruction-tuning loss is defined as: $\mathcal{L}(\theta) = -\frac{1}{\sum_{(x,y)\in\mathcal{D}} |y|} \sum_{(x,y)\in\mathcal{D}} \sum_{t=1}^{|y|} \log P_\theta\left(y_t | x, y_{<t}\right)$, this suggests that data selected by T-SHIRT provides more informative and challenging tokens for instruction tuning. We believe this observation provides meaningful evidence of token-level advantages.
>
> ### **W2: Distribution and impact of low $|\Delta_t|$ values.**
> We thank the reviewer for raising thoughtful questions about our decision to remove tokens with low $|\Delta_t|$ when computing S-IFD.
>
> **Cumulative effect of small $|\Delta_t|$ values.** First, we clarify that tokens with small $|\Delta_t|$ values can still have a significant cumulative effect. For instance, when using a token selection ratio of $k$% $= 75$%, we remove the bottom 25% of tokens (those with $|\Delta_t| \le 0.013$) from the S-IFD calculation. For a relatively long input, a lot of tokens are less informative and have low $|\Delta_t|$, but cumulatively, they could still add up. As illustrated in Figure 2 in the draft, we can differentiate between two samples with nearly identical IFD scores (0.998 vs. 0.998) effectively by their S-IFD scores (0.751 vs. 0.968).
>
> **Distribution of $|\Delta_t|$.** Second, since the rebuttal guideline does not allow presenting figures, we include a table to show the distribution of $|\Delta_t|$ values computed by *GPT-2* across the Alpaca-GPT4 dataset.
> | Ratio            | $10$% | $20$% | $30$% | $40$% | $50$% | $60$% | $70$% | $80$% | $90$% |
> | :--------------: |:-----: |:-----: |:-----: |:-----: |:-----: |:-----: |:-----: |:-----: |:-----: |
> | $\|\Delta_t\|$     | $\le 0.002$ | $\le 0.009$ | $\le 0.018$ | $\le 0.032$ | $\le 0.050$ | $\le 0.078$ | $\le 0.121$ | $\le 0.210$ | $\le 0.509$ |
>
> For example, 20% of the tokens have $|\Delta_t|$ below 0.009, and 50% of the tokens have $|\Delta_t|$ below 0.050. suggesting a considerable portion of tokens are uninformative.
>
> **Alternative weighting schemes.** Third, following the reviewer’s suggestion, based on Table 1, we conduct new experiments using *Qwen-2.5-7B* and Alpaca-GPT4 to explore alternative weighting schemes. In Equation (3), our default approach assigns a weight $w_t = 1$ to informative tokens and $w_t = 0$ to the rest. We compared this to two alternative schemes:
> * **Weights $1.5:1$.** $w_t = 1.5$ for informative tokens, $w_t = 1$ for uninformative tokens.
> * **Weights $2:1$.** $w_t = 2$ for informative tokens, $w_t = 1$ for uninformative tokens.
>
> The results are shown below.
> | Method | ARC‑C | HS | MMLU | TQA | BBH | GSM | $\mu_{\text{open}}$ |
> | :--------------------------- | :-----: | :-----: | :-----: | :-----: | :-----: | :-----: | :-----: |
> | **IFD** | $64.76$ | $\mathbf{82.66}$ | $\mathbf{74.33}$ | $60.86$ | $53.76$ | $\underline{86.50}$ | $70.48$ |
> | **T‑SHIRT ($k=50$, Weights $1.5:1$)** | $65.44$ | $82.37$ | $74.09$ | $\underline{61.44}$ | $53.83$ | $86.05$ | $70.54$ |
> | **T‑SHIRT ($k=50$, Weights $2:1$)** | $\underline{65.53}$ | $82.17$ | $74.19$ | $60.73$ | $\underline{53.95}$ | $\mathbf{86.81}$ | $\underline{70.56}$ |
> | **T‑SHIRT ($k=50$, Ours)** | $\mathbf{66.21}$ | $\underline{82.39}$ | $\underline{74.23}$ | $\mathbf{61.58}$ | $\mathbf{54.21}$ | $\mathbf{86.81}$ | $\mathbf{70.91}$ |
>
> Experimental results show that both weighting schemes outperform the IFD baseline on $\mu_{\text{open}}$, confirming the benefit of emphasizing more informative tokens during data selection. However, our default setting that removes uninformative tokens (i.e., setting $w_t = 0$ like Equation (3)) achieves even stronger performance, both on average and across all six individual benchmarks.
>
> These results validate our design choice. We will include the analysis and new experiment results in our revision.
>
> ### **Q1: Other scoring models.**
> ***GPT-2* vs. larger models.** In our early explorations, we considered several alternative scoring models and ultimately chose *GPT-2* for its strong efficiency and accessibility, as it can run quickly on a single consumer-grade GPU. While different scoring models can influence the quality of selected data, we refer the reviewer to the related work Superfiltering \[1\] for further information. Its Table 3 shows that larger models may offer only marginal improvements in data selection quality.
>
> **Open-source models vs. proprietary models (APIs).** Additionally, we chose open-source models over closed-source APIs because they provide access to token-level information and token embeddings, which are essential for our method. In contrast, querying more powerful proprietary LLMs via APIs typically returns only coarse-grained, categorical quality scores, as used in methods like DEITA \[2\] and DS$^2$ \[3\]. As shown in our Table 1, this lack of fine-grained information negatively affects the quality of selected data.
>
> **References**
>
> \[1\] M. Li, Y. Zhang, S. He, Z. Li, H. Zhao, J. Wang, N. Cheng, and T. Zhou. Superfiltering: Weak-to-Strong Data Filtering for Fast Instruction-Tuning, 2024.
>
> \[2\] W. Liu, W. Zeng, K. He, Y. Jiang, and J. He. What Makes Good Data for Alignment? A Comprehensive Study of Automatic Data Selection in Instruction Tuning, 2024.
>
> \[3\] J. Pang, J. Wei, A. Shah, Z. Zhu, Y. Wang, C. Qian, Y. Liu, Y. Bao, and W. Wei. Improving Data Efficiency via Curating LLM-Driven Rating Systems, 2025.

---

> ### Comment · Reviewer_MFfG · 2025-08-03
>
> Dear authors, thanks for your detailed response. All my concerns have been addressed and I have raised my score to 5.

---

> > ### Author Response · Authors · 2025-08-05
> >
> > Dear reviewer MFfG,
> >
> > Thank you for increasing your rating and for your thoughtful engagement throughout the review and discussion process! We are pleased to hear that our response addressed your concerns. Your comments and suggestions have been very helpful in improving our work, and we will incorporate the new results and clarifications into the next version of the paper.

---

### Official Review · Reviewer_wj2T · 2025-06-28

**Clarity:** 4
**Significance:** 3
**Originality:** 3
**Rating:** 5
**Confidence:** 4

**Summary:**

This paper introduces a novel framework for selecting high-quality instruction tuning data by addressing two important gaps in the existing literature: (1) overlooking token-level information for selection and (2) failing to incorporate robustness as a selection metric. The authors propose to firstly be selective when computing IFD and only consider the influence of tokens that exhibit large and informative signals, and secondly consider the neighborhood of samples through random perturbation in the embedding space and select data with high score and low variance. Through empirical experiments, the authors show impressive performance and efficiency of the proposed method.

**Questions:**

I don’t have much to complain about the paper. I am happy to have read a high-quality write-up with insightful methodology design and comprehensive experiments.

Some minor points:

1. It would be good to add repeated experiments and hence error bars for the experiments. Especially, baselines like RANDOM or DS2 have randomness in the selection, and the proposed T-SHIRT also uses random noises, it is critical to show the stability of the method through repeated, reproducible experiments. I could not be certain about the statistical significance right now.
2. It seems that the selection ratio $k$ is an essential hyperparameter that affects the performance much according to Table 1 and Figure 6. As your method does not rely on a validation dataset, what would be a recommended method to choose, or potentially optimize for this hyperparameter.
3. While the authors have included “Model Scale” as a limitation of the paper, the paper should at least show some larger-scale model experiments to show the scalability and generalizability of the proposed method. I suggest the authors to run these experiments.

**Ethical Concerns:**

["NO or VERY MINOR ethics concerns only"]

**Final Justification:**

I have nothing much to complain about this paper. For the suggestions raised during the rebuttal, the authors have provided sufficient clarification with additional results.

**Limitations:**

Yes

**Quality:**

3

**Strengths And Weaknesses:**

Strengths:

1. The paper builds on an interesting observation that not all tokens are contributing equal to the quality of the instruction tuning sample, hence proposing a novel method that carefully eliminates the effects of those less useful (or noisy) tokens when performing data selection.
2. The robustness perspective in scoring the utility of the data is new and useful.
3. Throughout the paper, there are good illustrative examples to demonstrate the rationale behind the specific designs of the methodology.
4. The proposed method exhibits impressive empirical performance, computational cost. At the same time, the paper includes extensive ablation studies.

For weaknesses, refer to the Questions section below.

---

> ### Author Rebuttal · Authors · 2025-07-30
>
> We thank the reviewer for the encouraging reviews on the technical novelty, empirical strength, and efficiency of our approach, as well as the quality of our writing. We also greatly appreciate the insightful feedback. Below, we provide detailed responses to each question (Q) raised by the reviewer.
>
> ### **Notations and clarifications.**
> In our response, we follow the draft’s notations for several benchmark names: ARC-C (ARC-Challenge), HS (HellaSwag), TQA (TruthfulQA), GSM (GSM8k), AH (Arena-Hard), and AE-2 (AlpacaEval 2.0). We use $\mu_{\text{open}}$ to denote the average performance across six OpenLLM Leaderboard benchmarks, $\mu_{\text{LLM}}$ for the average performance across two LLM-as-a-Judge benchmarks, and $\mu_{\text{all}}$ for the average performance over all eight benchmarks. Additionally, we clarify that *Gemini-2.5-Flash-Preview-04-17* was used in our draft as the judge for evaluating AH. Since this checkpoint is no longer available via API, we now use the stable version of *Gemini-2.5-Flash*, which affects the absolute scale of AH scores.
>
> ### **Q3: Applicability to larger models.**
> We thank the reviewer for this valuable suggestion. To address this concern, we use the settings presented in Table 2 and fine-tune a *Qwen-2.5-14B* model on data selected from the Magpie dataset. Here, we still use the token-selection ratio $k$% of 50% without further hyperparameter tuning because of time limitations. The evaluation results are shown below.
>
> | Method                      | ARC‑C |   HS   | MMLU |  TQA |  BBH |  GSM | $\mu_{\text{open}}$ |   AH  | AE‑2 | $\mu_{\text{LLM}}$ | $\mu_{\text{all}}$ |
> | :---------------------------|:-----: |:-----: |:-----: |:-----: |:-----: |:-----: |:-----: |:-----: |:-----: |:-----: |:-----: |
> | **Longest**                 | $68.17$ | $\underline{84.01}$ | $\underline{79.23}$ | $\mathbf{58.69}$ | $61.00$ | $\mathbf{85.22}$ | $\underline{72.72}$ | $\underline{34.40}$ | $35.01$ | $34.71$ | $\underline{63.22}$ |
> | **IFD**                     | $\mathbf{68.94}$ |	$\mathbf{84.05}$ | $79.07$ | $57.87$ | $\underline{61.78}$ | $83.02$ | $72.46$ | $33.10$ | $\underline{37.15}$ | $\underline{35.13}$ | $63.12$ |
> | **T‑SHIRT ($k=50$)**        | $\underline{68.60}$ | $83.90$ | $\mathbf{79.26}$ | $\underline{58.60}$ | $\mathbf{62.04}$ | $\underline{84.15}$ | $\mathbf{72.76}$ | $\mathbf{35.30}$ | $\mathbf{38.45}$ | $\mathbf{36.88}$ | $\mathbf{63.79}$ |
>
> Our method consistently surpasses the strongest baselines in terms of $\mu_{\text{open}}$, $\mu_{\text{LLM}}$, and $\mu_{\text{all}}$. In particular, it yields an average gain of at least 1.75 points on LLM-as-a-Judge benchmarks. These findings indicate that our data selection approach remains robust and advantageous for instruction tuning of models beyond the 7–8B parameter scale.
>
> We will include the new experiment in our revision.
>
> ### **Q1: Statistical significance of results for randomized methods.**
> We thank the reviewer for raising the importance of demonstrating the stability of our method, given the stochasticity in both T-SHIRT and certain baselines. We conduct repeated experiments on *Qwen-2.5-7B*, running both the Random baseline and our method (T-SHIRT with $k = 50$) three times with different random seeds. We report the mean and standard deviation of their performances across the OpenLLM Leaderboard benchmarks. Due to the budget constraints, we were unable to re-run DS$^2$ \[1\], since its randomness primarily stems from API queries, and re-scoring the dataset would require an expensive number of API queries. For context, we reiterate the performance for DS$^2$ and IFD from Table 1. Standard deviations for the Random baseline and T-SHIRT are shown in brackets.
>
> | Method                      | ARC‑C |   HS   |  MMLU |  TQA |  BBH |  GSM | $\mu_{\text{open}}$ |
> | :-------------------------- |:-----: |:-----: |:-----: |:-----: |:-----: |:-----: |:-----:|
> | **Random**                  | $64.42$ $(\pm 0.31)$ | $81.89$ $(\pm 0.07)$ | $74.15$ $(\pm 0.11)$ | $59.30$ $(\pm 0.18)$ | $53.46$ $(\pm 0.09)$ | $50.52$ $(\pm 14.17)$ | $63.96$ $(\pm 2.32)$ |
> | **DS**$^2$ | $65.10$ | $82.07$ | $\mathbf{74.35}$ | $60.58$ | $\underline{54.11}$ | $82.11$ | $69.72$ |
> | **IFD**                     | $64.76$ | $\mathbf{82.66}$ | $\underline{74.33}$ | $\underline{60.86}$ | $53.76$ | $\underline{86.50}$ | $\underline{70.48}$ |
> | **T‑SHIRT ($k=50$)**        | $\mathbf{66.04}$ $(\pm 0.24)$ | $\underline{82.36}$ $(\pm 0.03)$ | $74.09$ $(\pm 0.10)$ | $\mathbf{61.42}$ $(\pm 0.17)$ | $\mathbf{54.21}$ $(\pm 0.11)$ | $\mathbf{87.34}$ $(\pm 0.59)$ | $\mathbf{70.91}$ $(\pm 0.07)$ |
>
> These results show that our method consistently outperforms other baselines. Our method provides a considerable margin of 1.19 points over DS$^2$ on $\mu_{\text{open}}$, which makes it quite likely that this margin would remain robust, even under repeated trials for DS$^2$.
>
> We will include these statistical significance results in the revised version.
>
> ### **Q2: Selection and tuning of hyperparameter $k$.**
> We agree that the token-selection ratio $k$ used in S-IFD is a critical hyperparameter. We offer the following recommendations based on our findings:
>
> * **Default recommendation.** When resources are limited, we recommend starting with $k = 50$, which performs well across multiple models and datasets, as shown in both Table 1 and Table 2.
>
> * **Fine-grained tuning.** On *Llama-3.1-8B* with the Alpaca-GPT4 dataset, we conduct a sweep over $k \in \lbrace 50, 60, 70, 80, 90 \rbrace$. We find that our method works well across different values of $k$. The performance peaks at $k = 70$, with T-SHIRT outperforming IFD on 7 out of 8 benchmarks and remaining competitive on AH. It also achieves a 1.12-point improvement in $\mu_{\text{all}}$.
>
> | Method                      | ARC‑C |   HS   | MMLU |  TQA |  BBH |  GSM | $\mu_{\text{open}}$ |   AH  | AE‑2 | $\mu_{\text{LLM}}$ | $\mu_{\text{all}}$ |
> | :---------------------------|:-----: |:-----: |:-----: |:-----: |:-----: |:-----: |:-----: |:-----: |:-----: |:-----: |:-----: |
> | **IFD**                     | $61.35$ | $83.00$ | $62.88$ | $54.23$ | $46.99$ | $51.63$ | $60.01$ | $\mathbf{5.50}$ | $8.20$ | $6.85$ | $46.72$ |
> | **T‑SHIRT ($k=70$)**        | $\mathbf{63.48}$ | $\mathbf{83.26}$  | $\mathbf{63.92}$  | $\mathbf{55.19}$  | $\mathbf{47.13}$  | $\mathbf{53.15}$  | $\mathbf{61.02}$  | $5.40$ | $\mathbf{11.22}$  | $\mathbf{8.31}$  | $\mathbf{47.84}$  |
>
> These findings suggest that $k$ can be flexibly chosen within the range of 50–90. When tuning is feasible, it can provide additional gains.
>
> * **Transferability across model scales.** For the *Qwen-2.5* family, we find that $k = 50$ performs robustly across both 7B and 14B models, and across different instruction-tuning datasets (Alpaca-GPT4 and Magpie). We hypothesize that models within the same family share similar pretraining distributions, making it feasible to tune $k$ on a smaller model and transfer that setting to a larger one.
>
> We will include this discussion and guidance in the revision.

---

> > ### Comment · Reviewer_wj2T · 2025-08-04
> >
> > I would like to thank the authors for the additional results and clarifications during the rebuttal period. Please include them in the final version of the revised paper. I will keep my positive score.

---

> ### Author Response · Authors · 2025-08-05
>
> Dear reviewer wj2T,
>
> Thank you for your insightful feedback and for your active engagement during the discussion! We are very grateful for your helpful comments and suggestions, which have significantly contributed to improving our work. We will make sure to include the additional results and clarifications in the revised version of the paper.

---

### Official Review · Reviewer_6LJR · 2025-07-02

**Clarity:** 3
**Significance:** 3
**Originality:** 2
**Rating:** 4
**Confidence:** 4

**Summary:**

Many recent papers have studied how to select samples to use for instruction tuning by scoring with an LLM. In particular, an approach called IFD scores samples based on the change in response perplexity when conditioned on the instruction. However, this approach has two issues: 1) the change in response perplexity is computed over all tokens, even though some tokens in the response are very predictable regardless of the instruction (i.e., low change in PPL), and 2) the change in response perplexity may be sensitive to the choice of instruction --- small perturbations may impact this score. To address this, the paper presents T-SHIRT, which scores samples based on the top k% of token-level perplexity changes, rather than the perplexity change over the entire response. This scoring mechanism reduces the noise from uninformative tokens, like "Einstein" often coming after "Albert". Second, T-shirt more robustly selects samples based on a smoothed score within a neighborhood, where the neighborhood is constructed by adding Gaussian noise to the token embeddings. Experiments are conducted on two instruction tuning datasets and Llama/Qwen as base models.

**Questions:**

1. In Table 1/Table 2, there are some tasks (HS, MMLU, BBH, AH) where IFD does better than both versions of T-SHIRT. Does there exist a k besides 50%/75% that does allow for T-SHIRT to outperform IFD?
2. Can we better understand the inconsistency in the relationship between k% and the performance across Qwen and Llama? For instance, perhaps a more fine-grained sweep of k or an assessment on another LM would be helpful.
3. What does token embedding perturbation look like in terms of the actual text? Does it look vastly different from the word substitution example provided that swaps "mean"/"average"?

Overall, my score is a borderline accept. The proposed modifications are well-motivated and likely to be useful in practical, budget-constrained settings. However, I am not giving a 5 because I believe the experiments could be expanded to more thoroughly assess the limits of T-SHIRT and its behavior across different base models. I also find the novelty somewhat incremental, though I defer to reviewers with deeper familiarity in this area, and I am open to changing my opinion on this.

**Ethical Concerns:**

["NO or VERY MINOR ethics concerns only"]

**Limitations:**

Yes

**Quality:**

2

**Strengths And Weaknesses:**

**Strengths**

Quality:
- T-SHIRT's performance is demonstrated on both Alpaca-GPT-4 and Magpie datasets. This is promising, given that these datasets differ in how old they are and in terms of how heavily curated they are. This suggests that T-SHIRT is robust and may continue to offer improvements on future instruction tuning datasets.

Clarity:
- Paper is well-motivated, with illustrative examples of 1) cases where IFD and S-IFD have different vs similar scores and 2) cases where IFD/S-IFD are sensitive to the wording in the instruction.

Significance:
- The ability to score samples for data selection in a cheap and effective way is important. In comparison, many existing data selection methods involve using relatively large, costly LLMs.

**Weaknesses**

Quality:
- In Table 1/Table 2, there are some tasks (HS, MMLU, BBH, AH) where IFD does better than both versions of T-SHIRT. Is this due to the choice of T-SHIRT's hyperparameter k? That is, is there a setting of k where T-SHIRT consistently outperforms IFD, but this was not identified due to coarse sweeping? The method would be more compelling if you could confirm that there always is some k/other hyperparams such that using top-k% token scoring always outperforms scoring based on all tokens equally.
- In Figure 6a, it seems like the trends of k% vs performance are not consistent between Llama and Qwen. For Llama, higher k% translates to better performance, with approximately a 1 point difference between k=25 vs 75 - this is quite significant when you look at the range of accuracies in Table 1. For Qwen, performance appears to slightly decrease as k% increases. This inconsistency is surprising given that 28% of tokens have low $\triangle_t$ for both Llama and Qwen (L152). This raises concerns about whether we fully understand how T-SHIRT interacts with different base models, and about how practitioners using other base models should set k if they want to use this method.
- A minor suggestion to strengthen the case for T-SHIRT in comparison to methods that use API-based models to score samples: it would be interesting if one could implement DEITA/DS^2 using GPT2-based scores. Perhaps, you could show that GPT-2 fails to produce meaningful scores, and thus T-SHIRT/IFD are the only methods that would work at all when using a smaller LM. That is, can you argue that if a practitioner is budget constrained and must use a weaker LM, then no construction of DEITA or DS² will be viable? This would help clarify a practical niche that T-SHIRT/IFD has.

Clarity:
- What does token embedding perturbation look like in terms of the actual text? Does it look vastly different from the word substitution example provided that swaps "mean"/"average"?

Originality:
- While the proposed modifications to IFD are sensible, they have somewhat limited novelty, given that token-level scoring and neighborhood smoothing for robustness are well-known in many contexts. However, I am not completely familiar with all the works in the instruction tuning data selection space, so this concern is minor.

---

> ### Author Rebuttal · Authors · 2025-07-30
>
> We thank the reviewer for the positive assessment of our method’s scalability, well-motivated design, and efficiency, and we appreciate the constructive feedback. We hope the following clarifications help address the weaknesses (W) and questions (Q) raised by the reviewer.
>
> ### **Notations and clarifications.**
> In our responses, we follow the draft’s notations for several benchmark names: ARC-C (ARC-Challenge), HS (HellaSwag), TQA (TruthfulQA), GSM (GSM8k), AH (Arena-Hard), and AE-2 (AlpacaEval 2.0). We use $\mu_{\text{open}}$ to denote the average performance across six OpenLLM Leaderboard benchmarks, $\mu_{\text{LLM}}$ for the average performance across two LLM-as-a-Judge benchmarks, and $\mu_{\text{all}}$ for the average performance over all eight benchmarks. Additionally, we clarify that *Gemini-2.5-Flash-Preview-04-17* was used in our draft as the judge for evaluating AH. Since this checkpoint is no longer available via API, we now use the stable version of *Gemini-2.5-Flash*, which affects the absolute scale of AH scores.
>
> ### **W1 & Q1: Performance comparison between IFD and T-SHIRT across benchmarks.**
> We thank the reviewer for the thoughtful suggestion regarding finer hyperparameter tuning. In our draft, we used $k = 50$ and $k = 75$ to show that T-SHIRT can perform well even without heavy hyperparameter tuning. To explore hyperparameter choice further, we conduct a finer sweep over $k \in \lbrace 50, 60, 70, 80, 90 \rbrace$ using the same dataset and evaluation benchmarks from Table 1. We found that for *Llama-3.1-8B*, performance peaks at $k = 70$. We show the results in the following table.
>
> | Method                      | ARC‑C |   HS   | MMLU |  TQA |  BBH |  GSM | $\mu_{\text{open}}$ |   AH  | AE‑2 | $\mu_{\text{LLM}}$ | $\mu_{\text{all}}$ |
> | :---------------------------|:-----: |:-----: |:-----: |:-----: |:-----: |:-----: |:-----: |:-----: |:-----: |:-----: |:-----: |
> | **IFD**                     | $61.35$ | $83.00$ | $62.88$ | $54.23$ | $46.99$ | $51.63$ | $60.01$ | $\mathbf{5.50}$ | $8.20$ | $6.85$ | $46.72$ |
> | **T‑SHIRT ($k=70$)**        | $\mathbf{63.48}$ | $\mathbf{83.26}$  | $\mathbf{63.92}$  | $\mathbf{55.19}$  | $\mathbf{47.13}$  | $\mathbf{53.15}$  | $\mathbf{61.02}$  | $5.40$ | $\mathbf{11.22}$  | $\mathbf{8.31}$  | $\mathbf{47.84}$  |
>
> With $k = 70$, T-SHIRT outperforms IFD on 7 out of 8 benchmarks and performs competitively on the remaining one (AH). Specifically, T-SHIRT improves over IFD by:
> * +1.01 points on $\mu_{\text{open}}$,
> * +1.46 points on $\mu_{\text{LLM}}$,
> * +1.12 points on $\mu_{\text{all}}$.
>
> These results confirm that with appropriate tuning of $k$, T-SHIRT can consistently outperform IFD. We will include these results in the revised version.
>
>
> ### **W2 & Q2: Different choices of hyperparameter $k$ for *Llama* and *Qwen*.**
> **Clarification on the token selection ratio $k$%.** We clarify that, in Equation (3), we include a token $y_t$ in the computation of S-IFD only if its token-level informativeness $|\Delta_t|$ ranks in the top $k$% of the dataset. This means that we filter out the least informative $(100-k)$% of tokens, which are often easy to predict and potentially noisy.
>
> **Interpretation for Figure 6a.** The trends in Figure 6a show that:
> * On *Llama-3.1-8B*, using $k=75$, i.e., removing 25% of the least informative tokens from S-IFD computation, yields strong performance. In contrast, setting $k=25$, which removes 75% of the tokens, is too aggressive and degrades performance.
> * On *Qwen-2.5-7B*, we observe the best performance at $k=50$, indicating that this stronger pre-trained model can benefit from filtering out a larger fraction (50%) of the least informative tokens.
>
> These trends align with our pilot analysis (L151–L152 in our draft), which indicates that at least 22–28% of tokens are uninformative and can be excluded from S-IFD computation. We will include our clarification and additional results (e.g., for $k = 70$ that removes 30% of the least informative tokens) in the revision.
>
> ### **W4 & Q3: Token embedding perturbations and word substitutions.**
> **The random noise is continuous.** The noise injected into token embeddings is continuous and cannot be directly mapped back to discrete tokens, unlike word substitutions (e.g., replacing "mean" with "average"). By choosing an appropriate noise level, the perturbed embedding remains close to the original token embedding in the embedding space.
>
> **Intention of choosing token embedding perturbations.** We clarify that we deliberately designed embedding perturbation as a computationally efficient alternative to simulating lexical perturbations such as word substitutions. Below, we provide further discussion of our rationale for choosing continuous embedding perturbation:
> * **Utility.** Rule-based substitutions may struggle to preserve semantic equivalence and fluency. Prior works \[1, 2\] require costly human evaluations to verify the utility.
> * **Efficiency.** Prompting APIs to generate perturbed variants is prohibitively expensive. For example, perturbing each of the 52k Alpaca-GPT4 samples 30 times would require over 1.5 million API queries.
>
> Therefore, embedding perturbation is a lightweight, efficient way to consider robustness in scoring.
>
> We will clarify this distinction and rationale more clearly in the revision.
>
> **References**
>
> \[1\] D. Jin, Z. Jin, J. T. Zhou, and P. Szolovits. Is BERT Really Robust? A Strong Baseline for Natural Language Attack on Text Classification and Entailment, 2020.
>
> \[2\] B. Wang, C. Xu, S. Wang, Z. Gan, Y. Cheng, J. Gao, A. H. Awadallah, and B. Li. Adversarial GLUE: A Multi-Task Benchmark for Robustness Evaluation of Language Models, 2021.
>
> ### **W3: Feasibility of DEITA/DS$^2$ with smaller models (e.g., *GPT-2*).**
> We thank the reviewer for this thoughtful suggestion. It inspires us to reflect on how T-SHIRT compares with methods that rely on API-based scoring.
>
> **Discussion on DEITA.** For DEITA, two critical components depend on powerful LLMs like *ChatGPT*:
> * **Dataset evolution.** DEITA uses the API to generate increasingly complex and higher-quality instruction data following WizardLM \[1\]. *GPT-2* is not capable of performing this evolution.
> * **Scoring and supervision.** DEITA prompts *ChatGPT* to score each evolved sample, then uses these scores as supervision to train scorer LLMs. *GPT-2* is not suitable for this direct scoring via prompting. Using *GPT-2* would likely require fine-tuning it to serve as a weak classifier.
>
> **Discussion on DS$^2$.** DS$^2$ relies directly on prompting large LLMs for sample scoring, so it would likewise be infeasible to use *GPT-2* as a substitution.
>
> We believe this discussion illustrates a practical advantage of our method and will include this comparison in the revision.
>
> **References**
>
> \[1\] C. Xu, Q. Sun, K. Zheng, X. Geng, P. Zhao, J. Feng, C. Tao, Q. Lin, and D. Jiang. WizardLM: Empowering Large Pre-Trained Language Models to Follow Complex Instructions, 2024.
>
> ### **W5: Novelty of our method.**
> We acknowledge that token-level scoring and neighborhood smoothing for robustness have been explored in other contexts. We have reviewed related work in other contexts in Section 1.1 of our draft. However, to the best of our knowledge, we are the first to question the well-accepted paradigm in instruction tuning data selection, where each sample is scored as a whole unit and selected via a hard threshold. One of the key contributions of our work is a novel perspective: combining fine-grained token-level informativeness with neighborhood-level robustness for data selection. Our approach offers new insights into what makes a data point valuable for instruction tuning.

---

> > ### Comment · Reviewer_6LJR · 2025-08-05
> >
> > Thank you for your response!
> >
> > W2/Q2: If I understand correctly, you are saying that performance in k is expected to follow a parabola---k=100% keeps everything, including uninformative tokens, so decreasing k from 100 will help. However, if you decrease k too much, you start to score samples based on too few informative tokens, discarding valuable information. I think that extending Figure 6a to even k=10% could be helpful in showing this behavior (since for Qwen the trend looks a bit flat to me), and it would be good to add discussion about the tradeoffs going on as we vary these hyperparameters.
> >
> > > this stronger pre-trained model can benefit from filtering out a larger fraction (50%) of the least informative tokens.
> >
> > I do not completely understand why a stronger model would have smaller optimal k; can you explain more?
> >
> > W3: this is great---it really shows that these other approaches hinge on a performant LLM.

---

> > > ### Author Response · Authors · 2025-08-05
> > >
> > > ### **Follow-up on W2 & Q2**
> > > Thank you for your thoughtful and constructive feedback!
> > > > If I understand correctly, you are saying that performance in k is expected to follow a parabola.
> > >
> > > Yes, we expect the trend shown in Figure 6a to follow a parabolic pattern with respect to the token selection ratio $k$%.
> > >
> > > > I think that extending Figure 6a to even k=10% could be helpful in showing this behavior (since for Qwen the trend looks a bit flat to me), and it would be good to add discussion about the tradeoffs going on as we vary these hyperparameters.
> > >
> > > Thank you for this insightful suggestion! Following your advice, we set the token selection ratio to $k$% $= 10$% and applied our method to *Qwen-2.5-7B*. We observe that $\mu_{\text{open}}$ is 69.67, which is significantly lower than the value of 70.84 observed when $k$% $= 25$%. This further demonstrates that reducing $k$ too much can prune an excessive number of tokens during quality score computation for data selection, leading to the loss of valuable information. We will include this discussion about the trade-off involved in choosing the value of $k$ in our revised version.
> > >
> > > > I do not completely understand why a stronger model would have smaller optimal k; can you explain more?
> > >
> > > We hypothesize that the optimal boundary between informative and uninformative tokens (i.e., the choice of $k$) depends on the quality of the LLM's pretraining. A stronger pretrained model may benefit from being more selective and pruning a greater number of uninformative tokens during instruction tuning data selection. For instance, the *Llama-3.1* series \[1\] is pretrained on 15T tokens, while the *Qwen-2.5* series \[2\] is pretrained on 18T tokens, with differences in pretraining corpus distribution and quality. Given that *Qwen-2.5* is likely better pretrained, it may benefit from a smaller $k$ (e.g., $k=50$). A rigorous study of how pretraining quality influences the optimal choice of $k$ presents an exciting direction for future work.
> > >
> > > **References**
> > >
> > > \[1\] A. Grattafiori, A. Dubey, A. Jauhri, A. Pandey, A. Kadian, A. Al-Dahle, A. Letman, A. Mathur, A. Schelten, A. Vaughan, et al. The Llama 3 Herd of Models, 2024.
> > >
> > > \[2\] A. Yang, B. Yang, B. Zhang, B. Hui, B. Zheng, B. Yu, C. Li, D. Liu, F. Huang, H. Wei, et al. Qwen2.5 Technical Report, 2025.

---

> > > > ### Comment · Reviewer_6LJR · 2025-08-06
> > > >
> > > > Thank you for answering my questions! I will maintain my positive score.

---

> > > > > ### Author Response · Authors · 2025-08-07
> > > > >
> > > > > Dear Reviewer 6LJR,
> > > > >
> > > > > Thank you for taking the time to review our work and for actively engaging in the discussion! Your thoughtful feedback has been extremely helpful in improving the quality and clarity of our paper. We will incorporate the new results and clarifications into the next version of the work.

---

### Official Review · Reviewer_WDFx · 2025-07-05

**Clarity:** 3
**Significance:** 3
**Originality:** 3
**Rating:** 4
**Confidence:** 3

**Summary:**

This paper proposes a new instruction tuning method, called T-SHIRT, i.e., Token-Selective Hierarchical Data Selection for Instruction Tuning.  The general idea is to select high–quality instruction-tuning data to improve training efficiency and reduce data redundancy.  Some recent studies use LLM-based scoring functions, e.g., Instruction-Following Difficulty (IFD). While these data selection methods often lead to models that can match or even exceed the performance of models trained on the full datasets, this work identifies two key limitations: (i) they assess quality at the sample level, ignoring token-level informativeness; and (ii) they overlook the robustness of the scoring method, often selecting a sample due to superficial lexical features instead of its true quality.

This work proposes Token-Selective HIeRarchical Data Selection for Instruction Tuning (T-SHIRT), a novel data selection framework that introduces a new scoring method to include only informative tokens in quality evaluation and also promote robust and reliable samples whose neighbors also show high quality with less local inconsistencies. The paper demonstrates that models instruction-tuned on a curated dataset (only 5% of the original size) using T-SHIRT can outperform those trained on the entire large-scale dataset by up to 5.48 points on average across eight benchmarks.

**Questions:**

I raised a few questions in the weakness section.  Please check those questions and have some clear explanation.

**Ethical Concerns:**

["NO or VERY MINOR ethics concerns only"]

**Final Justification:**

Based on my reading of authors' rebuttals and other reviewers' comments, I would like to maintain my current evaluation comments and rating.

**Limitations:**

The limitation section discussed a few issues related to the current work.  I think the limitation discussion section is clear and informative.

**Paper Formatting Concerns:**

No formatting concerns observed.

**Quality:**

3

**Strengths And Weaknesses:**

1.  It identifies two limitations of recent instruction tuning works which use LLM-based scoring functions: (i) they assess quality at the sample level, ignoring token-level informativeness; and (ii) they overlook the robustness of the scoring method, often selecting a sample due to superficial lexical features instead of its true quality.

2. It proposes a new instruction tuning method, T-SHIRT, which selects high–quality instruction-tuning data to improve training efficiency and reduce data redundancy.  The new methodology on instruction tuning: It uses hierarchical selection to choose samples whose neighbors exhibit high average S-IFD and low variance. T heir token-selective hierarchical data selection for instruction tuning method introduces a new scoring method to include only informative tokens in quality evaluation and also promote robust and reliable samples whose neighbors also show high quality with less local inconsistencies

3.  High performance: The new method T-SHIRT can generate new instruction-tuned models on a much small, curated dataset (5% of the whole set) but outperforms those trained on the entire large-scale dataset.

4.  The overall analysis seems to be clear and in high quality.

 Weakness:

1. For Table 1 (the major performance comparison between our method, T-SHIRT, and other baseline methods on Alpaca-GPT-4), the performance gain seems not that great.   Especially using LLM as a judge (which is closer to human evaluation), the other method seems to show higher performance on Llama 3.1-8B.   I am not sure how you can explain this result.

2. The ablation study does not show quite impressive differences.  I am not sure how you can explain this performance result (Table 4).

3.  The experiments are all done on relatively smaller models (e.g., Llama 3.1-8B, Qwen 2.5-7B).  You mentioned this in the limitations but it should have some discussions on how the method can be extended to large models and what can be difficulties based on your experiments.

---

> ### Author Rebuttal · Authors · 2025-07-30
>
> We thank the reviewer for appreciating our technical contributions and high-quality analysis. We also thank the reviewer for the constructive comments. We hope the following responses address the weaknesses (W) raised by the reviewer.
>
> ### **Notations and clarifications.**
> In our responses, we follow the draft’s notations for several benchmark names: ARC-C (ARC-Challenge), HS (HellaSwag), TQA (TruthfulQA), GSM (GSM8k), AH (Arena-Hard), and AE-2 (AlpacaEval 2.0). We use $\mu_{\text{open}}$ to denote the average performance across six OpenLLM Leaderboard benchmarks, $\mu_{\text{LLM}}$ for the average performance across two LLM-as-a-Judge benchmarks, and $\mu_{\text{all}}$ for the average performance over all eight benchmarks. Additionally, we clarify that *Gemini-2.5-Flash-Preview-04-17* was used in our draft as the judge for evaluating AH. Since this checkpoint is no longer available via API, we now use the stable version of *Gemini-2.5-Flash*, which affects the absolute scale of AH scores.
>
> ### **W1: Performance gains of our method.**
>
> We believe this concern primarily stems from the hyperparameter settings used in our main experiments, particularly the token selection ratio $k$% defined in Equation (3). In the draft, we limited our evaluation to $k = 50$ and $k = 75$ to demonstrate that our method performs well even without extensive hyperparameter tuning.
>
> To address this concern, we conducted a finer-grained sweep over $k \in \lbrace 50, 60, 70, 80, 90 \rbrace$, using the same instruction tuning dataset and benchmark setup as in Table 1. On *Llama-3.1-8B*, performance peaks at $k = 70$. The full results are provided below.
>
> | Method                      | ARC‑C |   HS   | MMLU |  TQA |  BBH |  GSM | $\mu_{\text{open}}$ |   AH  | AE‑2 | $\mu_{\text{LLM}}$ | $\mu_{\text{all}}$ |
> | :---------------------------|:------: |:------: |:------: |:------: |:------: |:------: |:------: |:------: |:------: |:------: |:------: |
> | **IFD**                     | $61.35$ | $83.00$ | $62.88$ | $54.23$ | $46.99$ | $51.63$ | $60.01$ | $\mathbf{5.50}$ | $8.20$ | $6.85$ | $46.72$ |
> | **T‑SHIRT ($k=70$)**        | $\mathbf{63.48}$ | $\mathbf{83.26}$  | $\mathbf{63.92}$  | $\mathbf{55.19}$  | $\mathbf{47.13}$  | $\mathbf{53.15}$  | $\mathbf{61.02}$  | $5.40$ | $\mathbf{11.22}$  | $\mathbf{8.31}$  | $\mathbf{47.84}$  |
>
> With $k = 70$, our method, T-SHIRT, yields a clear improvement over the strongest baseline, IFD:
>
> * On $\mu_{\text{open}}$, T-SHIRT outperforms IFD by 1.01 points.
> * On $\mu_{\text{LLM}}$, which better approximates human preferences, T-SHIRT shows a 1.46-point gain.
> * Overall, T-SHIRT achieves a 1.12-point improvement in $\mu_{\text{all}}$.
> * T-SHIRT outperforms IFD on 7 out of 8 benchmarks and performs comparably on the remaining benchmark (AH).
>
> These results indicate that after tuning a single hyperparameter $k$, T-SHIRT consistently delivers stronger performance. We will include these results in the revised version.
>
> ### **W2: Interpretation of ablation study results.**
>
> **Both components bring improvements.** Our ablation study in Table 4 demonstrates that both of our proposed components, (1) S-IFD, which estimates token-level informativeness, and (2) hierarchical selection, which favors samples surrounded by consistently high-quality neighbors, contribute positively when applied independently. As shown in our response to W1, careful tuning of the hyperparameter $k$ further increases the performance gains of our method.
>
> **There is synergy between the two components.** While combining both components results in improved performance compared to using either one alone, the gains are not strictly additive. We believe this is due to the synergy between the components, as they may target partially overlapping aspects of data quality. We will clarify this interpretation in the revised version and plan to investigate their interaction more thoroughly in future work.
>
> ### **W3: Applicability to larger models.**
>
> We thank the reviewer for this valuable suggestion. To demonstrate the scalability of our method, we follow Table 2 and apply our approach (with $k = 50$) to the Magpie dataset and fine-tune a larger model, *Qwen-2.5-14B*. Due to time limitations, we do not conduct further hyperparameter tuning. The evaluation results are shown below:
> | Method                      | ARC‑C |   HS   | MMLU |  TQA |  BBH |  GSM | $\mu_{\text{open}}$ |   AH  | AE‑2 | $\mu_{\text{LLM}}$ | $\mu_{\text{all}}$ |
> | :---------------------------|:-----: |:-----: |:-----: |:-----: |:-----: |:-----: |:-----: |:-----: |:-----: |:-----: |:-----: |
> | **Longest**                 | $68.17$ | $\underline{84.01}$ | $\underline{79.23}$ | $\mathbf{58.69}$ | $61.00$ | $\mathbf{85.22}$ | $\underline{72.72}$ | $\underline{34.40}$ | $35.01$ | $34.71$ | $\underline{63.22}$ |
> | **IFD**                     | $\mathbf{68.94}$ |	$\mathbf{84.05}$ | $79.07$ | $57.87$ | $\underline{61.78}$ | $83.02$ | $72.46$ | $33.10$ | $\underline{37.15}$ | $\underline{35.13}$ | $63.12$ |
> | **T‑SHIRT ($k=50$)**        | $\underline{68.60}$ | $83.90$ | $\mathbf{79.26}$ | $\underline{58.60}$ | $\mathbf{62.04}$ | $\underline{84.15}$ | $\mathbf{72.76}$ | $\mathbf{35.30}$ | $\mathbf{38.45}$ | $\mathbf{36.88}$ | $\mathbf{63.79}$ |
>
> Our method consistently outperforms the strongest baselines across $\mu_{\text{open}}$, $\mu_{\text{LLM}}$, and $\mu_{\text{all}}$. Notably, it achieves an average improvement of at least 1.75 points on LLM-as-a-Judge benchmarks. These results suggest that our data selection method remains effective and beneficial for instruction tuning of models larger than 7-8B.
>
> We will include these results in the revised version.

---

> > ### Comment · Reviewer_WDFx · 2025-08-01
> >
> > I read the rebuttal and think the authors basically answered my questions.  I will maintain my original positive rating of the paper.   Thanks.

---

> ### Author Response · Authors · 2025-08-05
>
> Dear reviewer WDFx,
>
> We are glad that our rebuttal addressed your questions. Thank you for the time and effort you invested in reviewing our paper and engaging in the discussion! Your thoughtful feedback greatly helped us strengthen our work. We will incorporate the new results and clarifications into the revised version.

---

### Decision · Program_Chairs · 2025-09-17

**Decision:**

Accept (poster)

**Comment:**

This paper presents a new framework, T-SHIRT, for selecting high-quality instruction tuning data by addressing two key gaps in the existing literature: (1) the lack of token-level information in data selection, and (2) the absence of robustness as a selection criterion. T-SHIRT incorporates fine-grained token-level signals and prioritizes samples with consistently high-quality neighbors. The authors provide comprehensive experiments showing that datasets curated with T-SHIRT outperform both full datasets and those filtered by alternative selection methods.

The reviewers found the results to be novel, interesting, and relevant to the NeurIPS community. I therefore recommend acceptance.